# Loss of *Atoh1* from neurons regulating hypoxic and hypercapnic chemoresponses causes neonatal respiratory failure in mice

Meike E van der Heijden[1,2], Huda Y Zoghbi[1,2,3,4,5]*

[1]Department of Neuroscience, Baylor College of Medicine, Houston, United States; [2]Jan and Dan Duncan Neurological Research Institute, Texas Children's Hospital, Houston, United States; [3]Department of Molecular and Human Genetics, Baylor College of Medicine, Houston, United States; [4]Department of Pediatrics, Baylor College of Medicine, Houston, United States; [5]Howard Hughes Medical Institute, Baylor College of Medicine, Houston, United States

**Abstract** *Atoh1*-null mice die at birth from respiratory failure, but the precise cause has remained elusive. Loss of *Atoh1* from various components of the respiratory circuitry (e.g. the retrotrapezoid nucleus (RTN)) has so far produced at most 50% neonatal lethality. To identify other *Atoh1*-lineage neurons that contribute to postnatal survival, we examined parabrachial complex neurons derived from the rostral rhombic lip (rRL) and found that they are activated during respiratory chemochallenges. *Atoh1*-deletion from the rRL does not affect survival, but causes apneas and respiratory depression during hypoxia, likely due to loss of projections to the preBötzinger Complex and RTN. *Atoh1* thus promotes the development of the neural circuits governing hypoxic (rRL) and hypercapnic (RTN) chemoresponses, and combined loss of *Atoh1* from these regions causes fully penetrant neonatal lethality. This work underscores the importance of modulating respiratory rhythms in response to chemosensory information during early postnatal life.

DOI: https://doi.org/10.7554/eLife.38455.001

*For correspondence:
hzoghbi@bcm.edu

## Introduction

Hard-wired, transcriptionally defined neural circuit development is often complemented by synaptic plasticity that is driven by feedback from experience. Yet circuits giving rise to vital functions, such as respiration, have no time for such trial and error: the animal must be able to maintain its own $O_2$/$CO_2$ homeostasis from the moment it is born. This ability likely arises from a detailed genetic blueprint in the hindbrain respiratory circuit. Indeed, mapping the expression domains of key transcription factors in the developing hindbrain reveals a checkerboard pattern, with rostro-caudal and dorso-ventral stripes crisscrossing the entire region (*Gray, 2008, 2013*; *Pagliardini et al., 2008*; *Pasqualetti et al., 2007*). The complexity of the circuit and the relative inaccessibility of some its individual components have made it difficult to tease out the specific contributions of various neuronal populations to neonatal survival.

For example, mice lacking the transcription factor *Atonal homolog 1 (Atoh1)* die at birth of respiratory failure even though they generate some respiratory movements (*Ben-Arie et al., 1997*; *Rose et al., 2009b*; *Tupal et al., 2014*). The precise cause of this fully penetrant lethality has eluded a number of studies, which have nonetheless deepened our understanding of *Atoh1*'s contributions to the respiratory circuit. *Atoh1* is expressed in the developing hindbrain along the entire rostro-caudal rhombic lip, where it promotes the development of several nuclei involved in respiratory control (*Gray, 2013*; *Rose et al., 2009a, 2009b*; *Wang et al., 2005*). *Atoh1* is also expressed in

**eLife digest** Breathing seems very simple: humans and other animals do it all the time without even thinking about it. Yet, many different cell types coordinate rhythmic breathing movements. Some cells set the breathing rhythm, motor neurons control the muscles, and other cells sense blood oxygen and carbon dioxide levels. Information about oxygen and carbon dioxide is necessary to trigger faster and deeper breaths when there is too little oxygen, for example, at high altitude. Or when there is too much carbon dioxide, for example, during exercise.

At birth, most newborns can breathe as fast as needed because key genes oversee the development of all the cells involved in breathing. Learning more about these genes and what they do could lead to better understanding of why some newborns are at risk for sudden infant death or crib death. The Atoh1 gene, for example, helps carbon dioxide-sensing cells called retrotrapezoid neurons develop. Mice born without the Atoh1 gene are unable to breathe normally and die at birth. But when the gene is only deleted from these carbon dioxide-sensing cells in mice, just half of them die. This suggests that Atoh1 in other cells may also be important for breathing.

Now, Van der Heijden and Zoghbi show that the Atoh1 gene also helps develop another set of cells that are essential for breathing called the parabrachial complex. These cells receive information from oxygen sensors and relay the information to cells that set breathing rhythms. Mice missing parabrachial complex cells do not breathe faster when oxygen levels in the air are low. Mice lacking Atoh1 from both the parabrachial complex cells and the retrotrapezoid cells have breathing problems and die at birth.

Van der Heijden and Zoghbi show that the Atoh1 gene is essential for two cell types that make mice breathe faster when oxygen or carbon dioxide levels change. Together these two cell types are necessary for survival. The experiments also may provide insights into what goes wrong in babies who experience sudden infant death. Mutations in genes that are important to both cell types increase the risk of these infant deaths. Newborn babies with mutations in such key developmental genes will be at risk when in low oxygen or high carbon dioxide environments because their breathing systems are still maturing.

DOI: https://doi.org/10.7554/eLife.38455.002

postmitotic neurons of two paramotor nuclei: the intertrigeminal region (ITR) and the retrotrapezoid nucleus (RTN) (*Figure 1A and B*) (*Huang et al., 2012*; *Rose et al., 2009b*). The chemosensitive RTN sends excitatory projections to the preBötzinger complex (preBötC), which generates inspiratory rhythms (*Guyenet and Bayliss, 2015*; *Guyenet et al., 2010*; *Kumar et al., 2015*; *Onimaru and Homma, 2003*). *Atoh1* loss from the RTN impairs respiratory responses to hypercapnia, but, rather remarkably, causes only partial neonatal lethality (*Huang et al., 2012*).

We therefore set out to find other *Atoh1*-lineage neurons contributing to neonatal survival and further delineate *Atoh1*'s function in respiratory development using intersectional genetics. We found that loss of *Atoh1*-lineage neurons developing from the rostral rhombic lip (rRL) impairs both respiratory rhythm and chemoresponsiveness. Some of these neurons are specifically activated during respiratory chemoresponses and project to the *Atoh1*-lineage paramotor nuclei (ITR and RTN) as well as the preBötC. Yet only the combined deletion of *Atoh1* from the rRL and RTN recapitulated the fully penetrant lethality of *Atoh1*-null mice. This confirms that developmentally defined neural lineages have distinct roles in respiratory control and that, in neonatal mice, integration of chemosensory information is essential for survival.

## Results

### Rostral rhombic lip neurons are activated during chemochallenges

*Atoh1* is expressed along the entire rostro-caudal rhombic lip of the developing hindbrain (*Figure 1A*, red), where it functions as a proneural transcription factor. Loss of *Atoh1* results in loss of proliferating cells in the rhombic lip. Among the rhombic lip derived *Atoh1*-lineage are three populations of neurons that have been implicated in respiratory control: the parabrachial complex (PBC), the rostral ventral respiratory group (rVRG) and the lateral reticular nucleus (LRt) (*Figure 1B*, red)

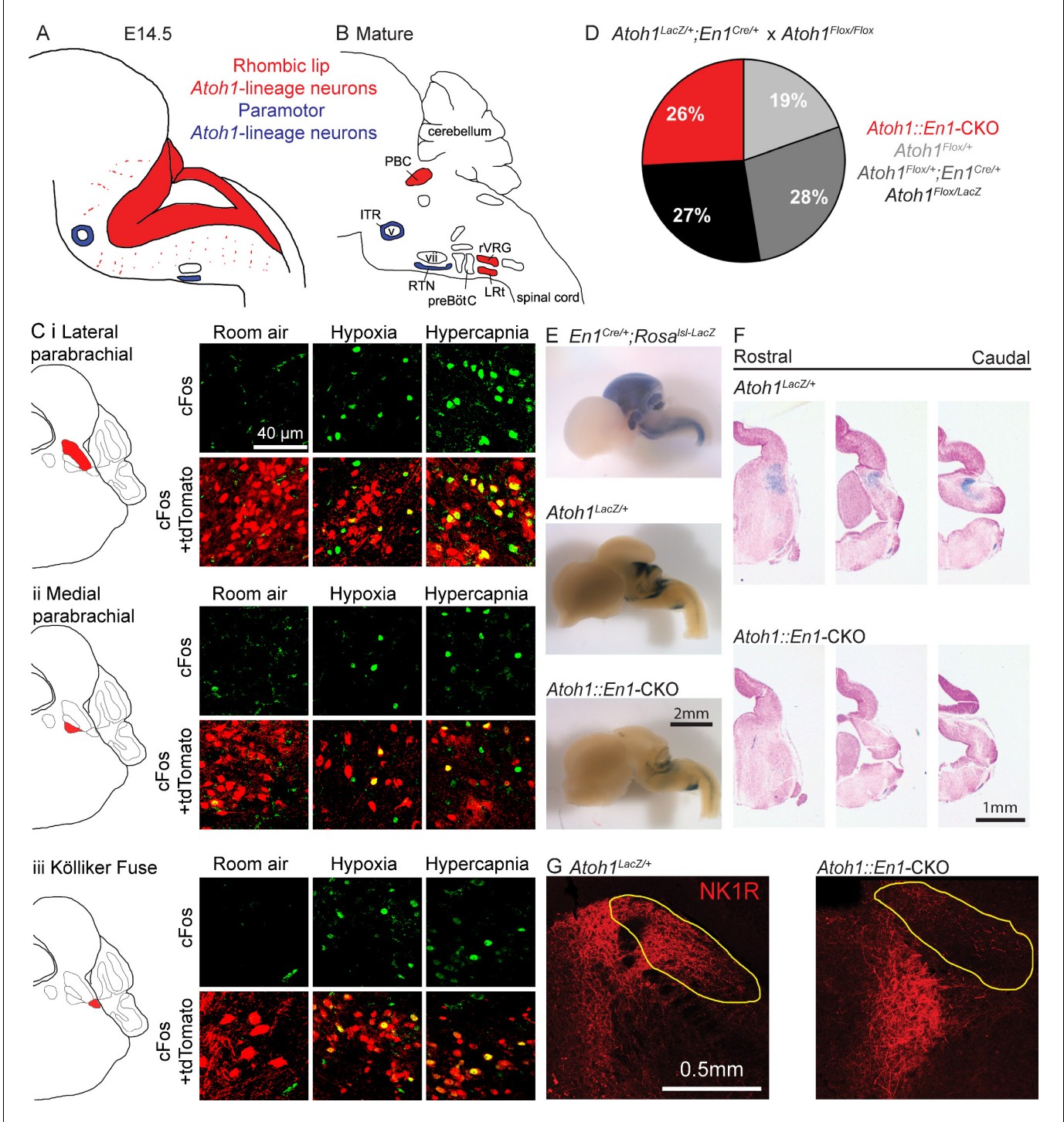

**Figure 1.** *Atoh1*-lineage parabrachial complex neurons are activated during chemochallenges but are not essential for neonatal survival. (A) Schematic of *Atoh1*-expression cells in the developing brainstem (E14.5). Red area represents proliferating cells in the rhombic lip. Blue cells are postmitotic neurons in paramotor nuclei. (B) Schematic of neural populations in the brainstem respiratory circuitry. Red nuclei represent rhombic lip, *Atoh1*-lineage neurons important for respiratory control. Blue nuclei are *Atoh1*-lineage neurons in two paramotor nuclei. (C) *Atoh1*-lineage neurons (tdTomato+) in three sub compartments of the parabrachial complex (i) lateral parabrachial, (ii) medial parabrachial and (iii) Kölliker Fuse) express the neural activity marker cFos selectively after a one-hour-exposure to hypoxia (10% $O_2$, balanced $N_2$) or hypercapnia (5% $CO_2$, 21% $O_2$, balanced $N_2$). (D) *Atoh1::En1*-CKO mice are born and survive in Mendelian ratios. (E) X-gal staining in *En1^{Cre/+};Rosa^{lsl-LacZ}* reporter allele, *Atoh1^{LacZ/+}* and *Atoh1::En1*-CKO E14.5 embryos to visualize *En1^{Cre}* expression and *Atoh1*-lineage cells in the developing brain. (F) Serial sections of X-gal-stained brains at the level of the pons in *Atoh1^{LacZ/+}* and *Atoh1::En1*-CKO mice. No *Atoh1*-lineage pontine PBC neurons develop in *Atoh1::En1*-CKO mice at E14.5. (G) Stain for NK1R

*Figure 1 continued*

receptor that is highly expressed in *Atoh1*-lineage PBC neurons. Loss of NK1R expression in *Atoh1::En1*-CKO mice at P21. Abbreviations: PBC, parabrachial complex; ITR, intertrigeminal region; RTN, retrotrapezoid nucleus; rVRG, rostral ventral respiratory group; LRt, lateral reticular; preBötC, preBötzinger complex; v, trigeminal motor nucleus; vii, facial motor nucleus.

DOI: https://doi.org/10.7554/eLife.38455.003

The following figure supplements are available for figure 1:

**Figure supplement 1.** *Atoh1*-lineage parabrachial complex neurons express specific markers.

DOI: https://doi.org/10.7554/eLife.38455.004

**Figure supplement 2.** *Atoh1*-lineage cerebellar neurons are not activated during chemochallenges.

DOI: https://doi.org/10.7554/eLife.38455.005

(*Rose et al., 2009b*; *Tupal et al., 2014*). Two paramotor nuclei express *Atoh1* during the postmitotic phase, and its expression is essential for their proper migration and connectivity from the RTN to the preBötC (*Figure 1A and B*, blue) (*Huang et al., 2012*; *Rose et al., 2009b*).

To date, the *Atoh1*-lineage PBC neurons are the only *Atoh1*-lineage neurons whose role in respiratory control and neonatal survival was not assessed. We first tested whether *Atoh1*-lineage PBC neurons might have a role in respiratory chemoresponses. To test this we labeled the *Atoh1*-lineage with tdTomato using a Cre-dependent reporter allele (*Atoh1^{Cre/+}*;*Rosa^{lsl-tdTomato/+}* mice) and exposed these mice to either room air, hypoxia, or hypercapnia prior to staining for the neural activity marker cFos. We found tdTomato$^+$, cFos$^+$ double-positive cells in the medial and lateral parabrachial region as well as in the Kölliker Fuse after exposure to either hypoxia or hypercapnia, but not room air (*Figure 1C*). This confirms that *Atoh1*-lineage PBC neurons are activated by changes in $O_2$ and $CO_2$ and might play a role in respiratory function.

There are no previous reports that PBC neurons are intrinsically chemosensitive, so these neurons are likely activated by upstream neurons that are chemosensitive. Previous studies showed that *Atoh1*-null mice lose the substance P receptor NK1R in the PBC region (*Rose et al., 2009b*). We found that indeed all NK1R-expressing PBC neurons were *Atoh1*-lineage neurons (*Figure 1—figure supplement 1A*). We also looked whether *Atoh1*-lineage PBC neurons expressed calcitonin gene-related peptide (CGRP) and pituitary adenylate cyclase-activating polypeptide (PACAP), because these peptides have been implicated in playing a role for hypercapnic and hypoxic responses respectively (*Arata et al., 2013*; *Cummings et al., 2004*; *Kaur et al., 2017*; *Yokota et al., 2015*). We found that all counted *Atoh1*-lineage neurons in the lateral PBC expressed CGRP and that some expressed PACAP (*Figure 1—figure supplement 1A*). Together, these results suggest that *Atoh1*-lineage neurons might be important for respiratory responses through signaling with one or both of these neuro peptides.

Next, we assessed whether these *Atoh1*-lineage neurons are required for neonatal survival. As PBC neurons develop from the rRL, we deleted *Atoh1* only from this domain using an *Engrailed-1*-driven Cre-line (*En1^{Cre}*). We crossed *Atoh1^{Flox/Flox}* females (*Shroyer et al., 2007*) to males heterozygous for *En1^{Cre/+}* and *Atoh1^{LacZ/+}*, which is a functional *Atoh1*-null allele that can be used to trace *Atoh1*-expressing neurons throughout development (*Ben-Arie et al., 2000*). *Atoh1^{Flox/LacZ}*;*En1^{Cre/+}* mice (hereafter *Atoh1::En1*-CKO) were born in Mendelian ratios (25/97 surviving pups, *Figure 1D*). We stained E14.5 embryos with X-gal and confirmed that *Atoh1::En1*-CKO lost all *Atoh1*-expressing neurons from the rRL, while leaving other *Atoh1*-lineage neurons intact (*Figure 1E and F*). Anatomical analysis of postnatal (P21) animals confirmed that this conditional deletion of *Atoh1* resulted in loss of NK1R-expressing PBC neurons (*Figure 1G*). Thus, despite losing these *Atoh1*-lineage PBC neurons, these animals survive, showing that *Atoh1* expression in the rRL is not necessary for neonatal survival.

*Atoh1::En1*-CKO mice developed severe ataxia, dystonia and tremor in the second to third week after birth and died shortly after weaning (P22-25), probably because the motor phenotypes impair their ability to get proper amounts of food and water. These phenotypes were likely the result of loss of *Atoh1*-lineage cerebellar neurons including glutamatergic deep cerebellar nuclei and cerebellar granule cells (*Figure 1—figure supplement 2A*) (*Ben-Arie et al., 1997*; *Wurst et al., 1994*). Unlike *Atoh1*-lineage PBC neurons, however, these *Atoh1*-lineage cerebellar neurons are not activated during respiratory chemochallenges and are thus less likely to be important for respiratory chemoresponses (*Figure 1—figure supplement 2B and C*).

## Rostral rhombic lip neurons contribute to respiratory rhythms

Given that *Atoh1::En1*-CKO mice survive the early neonatal period, we were able to examine their respiration using unrestrained whole-body plethysmography (UWBP) at three weeks of age (*Figure 2A*). In room air, *Atoh1::En1*-CKO mice had a greater number of sigh-induced and spontaneous apneas, sighs, and irregular respiratory rhythms than their control littermates (*Figure 2B*). No other respiratory parameters were affected (*Figure 2—figure supplement 1*).

Apneas, sighs, and rhythmic irregularity are hallmarks of immature respiration that can occur in some human infants (*Abu-Shaweesh and Martin, 2008*; *Martin et al., 2004*; *Abu-Shaweesh, 2004*). When infants present with apnea of prematurity (AOP) in a clinical setting, they are treated with

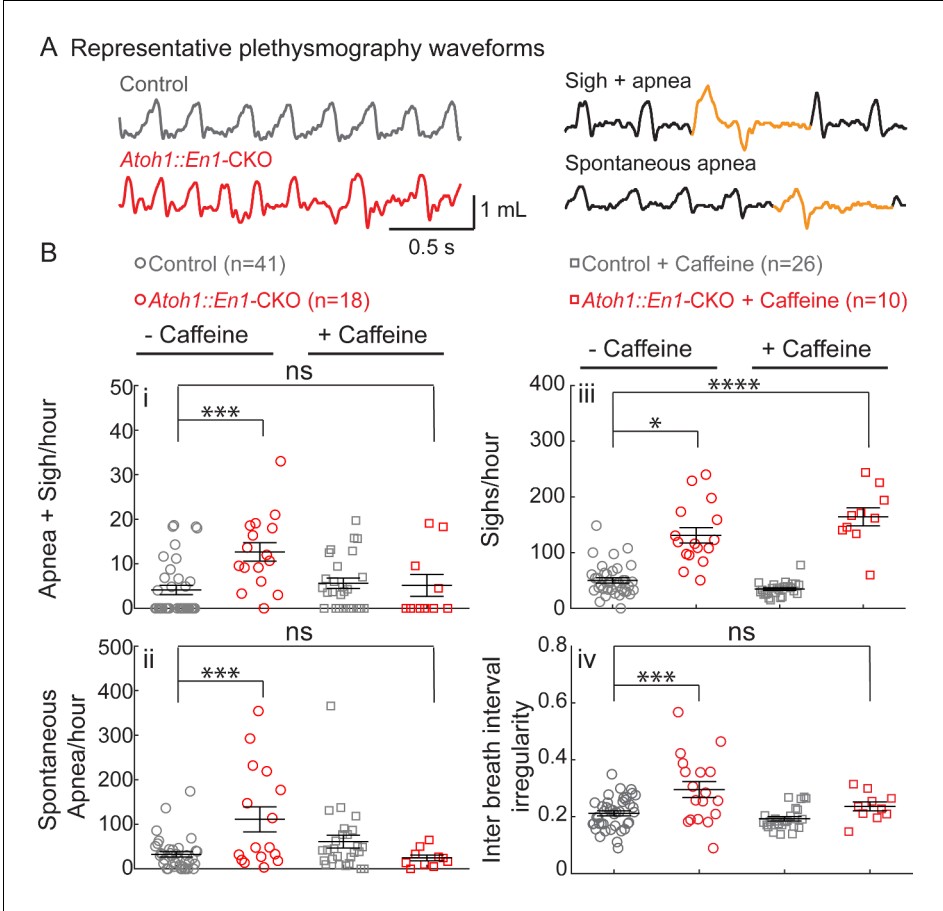

**Figure 2.** *Atoh1::En1*-CKO mice have many apneas causing irregular breathing rhythms that can be rescued by caffeine treatment in room air. (**A**) Representative plethysmography traces from a control and *Atoh1::En1*-CKO mouse. Example traces of apnea and sigh. (**Bi**) *Atoh1::En1*-CKO mice have more apneas following sighs and (**Bii**) spontaneous apneas per hour than control littermates. Caffeine treatment rescues apneas. (**Biii**) *Atoh1::En1*-CKO mice have more sighs per hour than control littermates, which was not be rescued by caffeine. (**Biv**) *Atoh1::En1*-CKO mice breathe more irregularly than control littermates, which can be rescued with caffeine treatment. Inter breath interval irregularity was defined as: absolute (breath length(n + 1) – breath length(n)/breath length(n). Significance was determined using a Two-way ANOVA (genotype*treatment), Tukey-Kramer post-hoc. *p<0.05. **p<0.01. ***p<0.001. ****p<0.0001. Error bars represent: mean ± SEM.

DOI: https://doi.org/10.7554/eLife.38455.006

The following source data and figure supplement are available for figure 2:

**Source data 1.** Raw plethysmography data room air recordings.
DOI: https://doi.org/10.7554/eLife.38455.008

**Figure supplement 1.** Caffeine treatment has opposing effects on changes in respiratory rhythms and tidal volume between *Atoh1::En1*-CKO mice and control littermates.
DOI: https://doi.org/10.7554/eLife.38455.007

caffeine to stabilize their breathing rhythms (*Aranda et al., 1977*; *Natarajan et al., 2007*). We therefore tested whether caffeine treatment could rescue respiratory rhythms in *Atoh1::En1*-CKO mice by administering caffeine through the drinking water of lactating dams from P2 onward. This was sufficient to detect caffeine levels in the blood plasma of the pups (treated: 6.15 ± 2.1 mg/L caffeine; untreated: 0.15 ± 0.04 mg/L caffeine; p=0.02, two-tailed t-test). These levels are similar to those observed in infants treated with caffeine (*Natarajan et al., 2007*).

Caffeine treatment normalized apnea frequency and irregular breathing rhythms, but not sighs (*Figure 2B*), showing that caffeine is sufficient to stabilize irregular breathing rhythms in our mice similar to human infants. Much to our surprise, this method of caffeine treatment also significantly decreased minute ventilation in control mice, as a result of both decreased tidal volume and breathing frequency (*Figure 2—figure supplement 1*). Although *Atoh1::En1*-CKO mice also showed a decrease in tidal volume, they seemingly compensated by increasing their rate of respiration, resulting in normal minute ventilation compared to control conditions.

## Rostral rhombic lip neurons are essential for respiratory chemoresponsiveness

As *Atoh1*-lineage PBC neurons are specifically activated during hypoxia and hypercapnia, we hypothesized that these neurons also contribute to respiratory chemoresponses. We therefore assessed whether *Atoh1::En1*-CKO mice showed abnormal responses to hypoxia and whether the caffeine treatment rescued their irregular respiratory rhythms by restoring proper chemoresponses. Control mice had a short initial increase in tidal volume and breathing frequency that returned to baseline within several minutes of exposure to hypoxia (*Figure 3A and B*). This resulted in a bimodal response in minute ventilation (*Figure 3C*), similar to what others have reported in juvenile mice (*Haddad and Mellins, 1984*; *Waters and Gozal, 2003*). *Atoh1::En1*-CKO mice also showed a brief initial increase in tidal volume (*Figure 3A*), but this was accompanied by a rapid, sustained decrease in respiratory rate that repressed minute ventilation during hypoxia (*Figure 3D*). Interestingly, while caffeine treatment in control mice did prolong the respiratory response to hypoxia by limiting respiratory depression during exposure to hypoxic gas (*Figure 3C*), it did not improve respiratory chemoresponses in *Atoh1::En1*-CKO mice (*Figure 3D*). Respiratory depression in response to hypoxia resembles the suppression of fetal breathing movements during hypoxia in prenatal mammals (*Gluckman and Johnston, 1987*; *Haddad and Mellins, 1984*; *Abu-Shaweesh, 2004*; *Waters and Gozal, 2003*), underscoring how loss of rRL neurons recapitulates many aspects of immature breathing control.

We next assessed how *Atoh1::En1*-CKO mice responded to hypercapnia. We found that control littermates showed a rapid increase in tidal volume and breathing frequency during hypercapnia, whereas respiratory chemoresponses of *Atoh1::En1*-CKO mice were severely attenuated (*Figure 3E and F*). Caffeine treatment delayed the return to baseline minute ventilation in control mice, but did not improve the hypercapnic chemoresponses of *Atoh1::En1*-CKO littermates (*Figure 3G and H*).

Thus, the mechanism by which caffeine rescues apneas, sighs, and irregular respiratory rhythms in *Atoh1::En1*-CKO mice, cannot be through normalizing chemoresponses and these results confirm that *Atoh1*-lineage rRL neurons contribute to respiratory chemoresponses to low oxygen and high carbon dioxide.

## The cerebellar cortex is not essential for respiratory chemoresponses

As neither *Atoh1*-lineage deep cerebellar nuclei nor cerebellar granule cells were activated during respiratory chemochallenges, we hypothesized that the abnormal chemoresponses, seen upon deletion of *Atoh1* from the rRL, were not due to cerebellar dysfunction. To date, there is no Cre-line that can specifically delete *Atoh1* from the pontine PBC without affecting the cerebellum, or vice versa. Therefore, we tested our hypothesis by silencing neurotransmission of Purkinje cells (PCs), which are the sole output of the cerebellar cortex. PCs receive input from granule cells and directly project onto deep cerebellar nuclei. These $Pcp2^{Cre/+};Slc32a1^{Flox/Flox}$ mice have no defects in cell types other than PCs, and loss of PC signaling results in abnormal motor control including ataxia and poor performance on the rotarod (*White et al., 2014*). Despite this abnormal cerebellar function, $Pcp2^{Cre/+};$ $Slc32a1^{Flox/Flox}$ mice do not have more apneas or sighs than control littermates during room air breathing, although they have slightly more irregular respiratory rhythms than their control

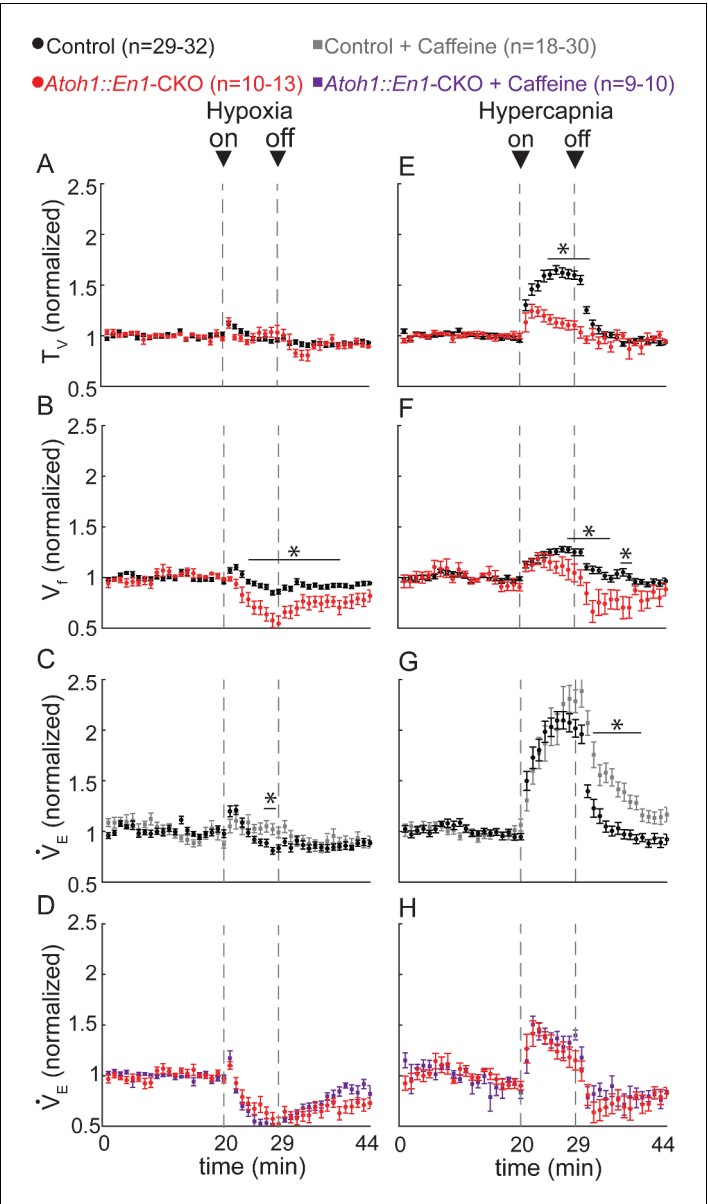

**Figure 3.** *Atoh1::En1*-CKO mice have abnormal respiratory chemoresponses that cannot be rescued by caffeine. Normalized changes in tidal volume ($T_V$) (**A**), respiratory frequency ($V_f$) (**B**), and minute ventilation ($V_E$) (**C** and **D**) during hypoxic challenge (10% $O_2$, balanced $N_2$). *Atoh1::En1*-CKO mice show respiratory repression during hypoxia (**B**) that cannot be rescued by caffeine (**D**). Normalized changes in $T_V$ (**E**), $V_f$ (**F**), and $V_E$ (**G** and **H**) during hypercapnic chemochallenge (5% $CO_2$, 21% $O_2$, balanced $N_2$). *Atoh1::En1*-CKO mice have attenuated response to hypercapnia (**E** and **F**) that cannot be rescued by caffeine (**H**). Significance was determined using a t-test (2-tailed) at each individual time point, *$p < 0.0011$ (0.05/44 for Bonferroni correction). Error bars represent mean ± SEM.
DOI: https://doi.org/10.7554/eLife.38455.009

The following source data and figure supplements are available for figure 3:

**Source data 1.** Raw plethysmography data respiratory chemoresponses.
DOI: https://doi.org/10.7554/eLife.38455.012

**Figure supplement 1.** Silencing cerebellar cortex output neurons does not cause abnormal respiratory control.
DOI: https://doi.org/10.7554/eLife.38455.010

**Figure supplement 1—source data 1.** Raw plethysmography data respiratory chemoresponses.
DOI: https://doi.org/10.7554/eLife.38455.011

littermates (*Figure 3—figure supplement 1Aiv*). No other respiratory parameters (*Figure 3—figure supplement 1A*) such as respiratory chemoresponses to either hypoxia or hypercapnia were altered upon silencing of PCs (*Figure 3—figure supplement 1B*). Thus, abnormal cerebellar function or ataxia does not explain the abnormal respiratory chemoresponses observed in *Atoh1::En1*-CKO mice.

## Rostral rhombic lip neurons are important for respiratory control in P7 mice

*Atoh1::En1*-CKO mice display irregular breathing rhythms, sighs, and apneas, as well as respiratory depression in response to hypoxia and attenuated respiratory response to hypercapnia. As noted above, these features characterize immature respiration in some human infants, although they usually resolve on their own with time (*Abu-Shaweesh and Martin, 2008*; *Abu-Shaweesh, 2004*). Our results suggest that rRL neurons might play a prominent role in the postnatal maturation of respiratory control.

To test this hypothesis, we evaluated respiratory control in one-week-old (P7) *Atoh1::En1*-CKO mice and control littermates (*Figure 4A*). At this age, *Atoh1::En1*-CKO mice display about twice as many apneas as their control littermates, but do not sigh more (*Figure 4B*). We also observed that P7 *Atoh1::En1*-CKO mice have longer inhalation times and shallower breaths, resulting in a slower breathing rhythm and smaller minute ventilation (*Figure 4B*).

In response to hypoxia, control mice showed the bimodal respiratory response we observed in P21 *Atoh1::En1*-CKO mice with an initial increase in tidal volume and respiratory frequency, followed by respiratory depression (*Figure 4Ci to Ciii*). Nevertheless, the decrease in tidal volume and respiratory frequency was initiated earlier during the hypoxic exposure and was more pronounced in *Atoh1::En1*-CKO mice. In contrast, there were no differences in respiratory response to hypercapnia between *Atoh1::En1*-CKO mice and control littermates at P7 (*Figure 4Civ to Cvi*).

Additionally, these results underscore our findings that cerebellar dysfunction is not the main driver of respiratory abnormalities in *Atoh1::En1*-CKO mice, the cerebellum is not yet developed at this age: cerebellar granule cells do not form their first functional synapses with Purkinje cells until P8 (*White and Sillitoe, 2013*). Thus, at P7 wild-type cerebelli are functionally more similar to those of *Atoh1::En1*-CKO mice, and if the cerebellum caused respiratory dysfunction in *Atoh1::En1*-CKO mice we would expect the differences between P7 control mice and *Atoh1::En1*-CKO mice to be smaller, not larger.

Our results show that P7 *Atoh1::En1*-CKO mice have abnormal respiratory control in room air and in response to hypoxia. This suggests that although some respiratory phenotypes of P21 *Atoh1::En1*-CKO mice (high number of sighs and abnormal hypercapnic chemoresponses) might indeed result from abnormal postnatal maturation, others (slow breathing rhythms and small tidal volume) might be self-resolving with maturation. Yet at all ages tested rRL neurons are essential to prevent apneas and respiratory depression during hypoxia.

## Rostral rhombic lip neurons project to *Atoh1*-lineage paramotor nuclei and the preBötC

Oxygen sensors in the carotid body communicate with the nucleus tractus solitarius (NTS) to integrate the sensory information into the central respiratory circuit, which is necessary for hypoxic chemoresponses (*Accorsi-Mendonça et al., 2015*; *Dutschmann et al., 2008*; *Ferreira et al., 2015*; *Mayer et al., 2015*; *Song et al., 2011*). There is ample evidence that NTS neurons directly project to the PBC (*Bianchi et al., 1995*; *Dutschmann et al., 2008*; *Roman et al., 2016*; *Song et al., 2011*) and our results suggest that the *Atoh1*-lineage PBC neurons are necessary to prevent respiratory depression in response to hypoxia. Yet it is unknown whether they modulate respiratory rhythms through activation of downstream rhythmogenic or chemosensitive nuclei, or act directly as premotor neurons.

To trace the projections from the *Atoh1*-lineage rRL neurons, we made use of an intersectional reporter allele (Ai65) that expresses tdTomato only after removal of both an FRT-flanked and a *loxP*-flanked stop-cassette (*Madisen et al., 2015*). We generated an *Atoh1^FlpO^* knock-in mouse line that expresses FlpO recombinase in place of Atoh1 under the Atoh1 promoter (*Figure 5—figure supplement 1*). This mouse line can be used to remove the first stop-cassette in the reporter allele,

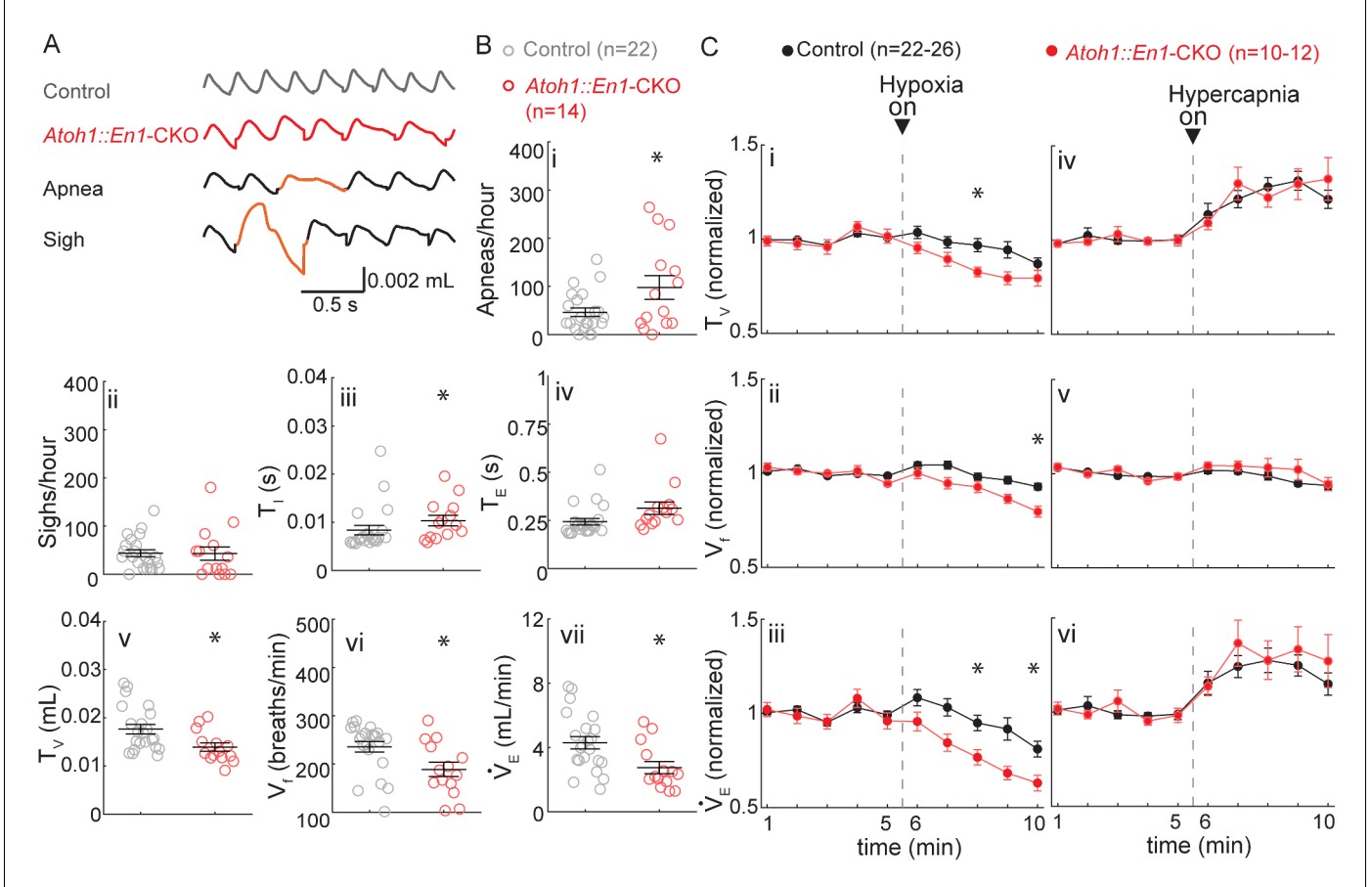

**Figure 4.** P7 *Atoh1::En1*-CKO mice have abnormal respiratory control. (**A**) Representative plethysmography traces from a control and *Atoh1::En1*-CKO mouse. Example traces of apnea and sigh. (**B**) One-week-old *Atoh1::En1*-CKO mice have significantly more apneas (i), longer inspiratory time ($T_I$) (iii), smaller tidal volume ($T_V$) (**v**), slower respiratory rhythms ($V_f$) (vi), and lower minute ventilation ($V_E$) (vii). Number of sighs (ii) and expiratory time ($T_E$) (iv) were not different. Significance for room air breathing parameters were determined using a t-test (2-tailed). *$p < 0.05$. (**C**) One-week-old *Atoh1::En1*-CKO mice have enhanced respiratory depression in response to hypoxia through an enhanced decrease in $T_V$ (i) and $V_f$ (ii), resulting in a decreased $V_E$(iii). There were no significant differences in the respiratory chemoresponses to hypercapnia (iv to vi). Significance was determined using a t-test (2-tailed) at each individual time point, *$p < 0.01$. Error bars represent mean ± SEM.

DOI: https://doi.org/10.7554/eLife.38455.013

The following source data is available for figure 4:

**Source data 1.** Raw data P7 plethysmography recordings.
DOI: https://doi.org/10.7554/eLife.38455.014

exclusively in *Atoh1*-lineage neurons (*Figure 5A*); the second stop-cassette is removed using *En1*[Cre], so that only neurons in the *Atoh1;En1* intersectional domain will be labeled with tdTomato (*Figure 5A*). We confirmed that the cell bodies of *Atoh1*-lineage parabrachial neurons were labeled red (*Figure 5B*) and tdTomato[+] puncta overlapped with the synaptic marker synapsin, thus representing synapses on downstream neurons (*Figure 5C*). We then assessed whether any tdTomato[+] puncta were found in key respiratory nuclei or motor nuclei involved in respiratory motor rhythms (Summarized in *Figure 5D*). We found tdTomato[+] puncta in the *Atoh1*-lineage paramotor nuclei: the intertrigeminal region (ITR) and the chemosensitive retrotrapezoid nucleus (RTN) (*Figure 5E and F*). We also found projections towards the rhythmogenic preBötC (*Figure 5G*), but detected no feedback projections to the nucleus tractus solitarius (NTS) (*Figure 5H*). We found no tdTomato[+] puncta in any of the motor nuclei in the respiratory circuitry (*Figure 5I–L*). This shows that the rRL neurons selectively innervate downstream neurons in the respiratory circuit that are important for chemoresponsive adaptations in the respiratory rhythm, but do not function directly as premotor neurons.

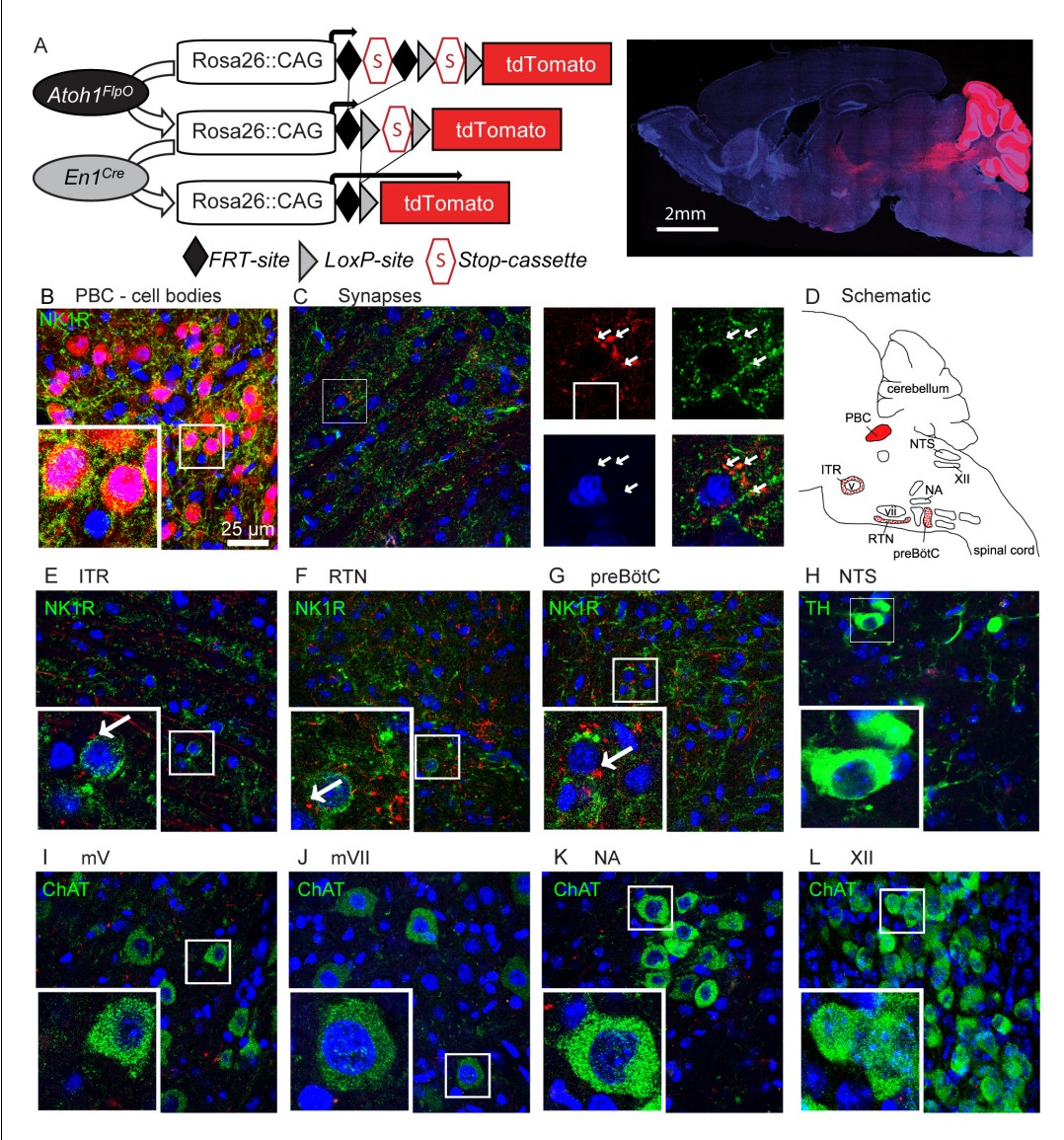

**Figure 5.** Rostral rhombic lip neurons project to paramotor nuclei and the preBötC. (**A**) Intersectional strategy to label only those neurons with a history of *Atoh1* (FlpO) and *En1* (Cre) expression. Representative whole-brain image of P21 *Atoh1^FlpO^;En1^Cre^;Ai65/+* mouse on the right. (**B**) Cell bodies in the PBC are tdTomato+. (**C**) tdTomato+ puncta overlap with the synapse-marker synapsin. (**D**) Schematic of the respiratory circuitry. Solid red PBC is the only respiratory nucleus expressing tdTomato. Dotted nuclei are the nuclei in which tdTomato+ projections were observed. TdTomato+ puncta near cell bodies were observed in the (**E**) ITR, (**F**) RTN, and (**G**) preBötC. No tdTomato+puncta were observed in the (**H**) NTS, or in respiratory motor nuclei (**I**) mV, (**J**) mVII, (**K**) NA, (**L**) XII. Abbreviations: PBC, parabrachial complex; ITR, intertrigeminal region; RTN, retrotrapezoid nucleus; preBötC, preBötzinger Complex; NTS, nucleus tractus solitarius; mV, trigeminal motor nucleus; mVII, facial motor nucleus; NA, nucleus ambiguus; XII, hypoglossal motor nucleus.

DOI: https://doi.org/10.7554/eLife.38455.015

The following figure supplement is available for figure 5:

**Figure supplement 1.** Generation of *Atoh1^FlpO^* mice.

DOI: https://doi.org/10.7554/eLife.38455.016

Loss of activation from the PBC might subdue the respiratory responses, impairing adaptation to hypercapnia and hypoxia.

### *Atoh1*-lineage intertrigeminal region neurons are not required for neonatal survival or normal respiratory chemoresponses

We found that rRL neurons project to the *Atoh1*-lineage intertrigeminal region (ITR) (*Figure 5E*), suggesting that part of the *Atoh1::En1*-CKO phenotype might be through loss of modulation of the ITR region. Loss of *Atoh1* from the ITR was previously assessed only in *Atoh1::Phox2b*-CKO mice that also have impaired RTN development (*Huang et al., 2012*; *Ruffault et al., 2015*).

To assess the function of *Atoh1*-lineage ITR neurons, we generated an ITR-specific conditional knockout mouse using the *HoxA2::Cre*^TG^ mouse line that expresses Cre only in r2 neurons (*Awatramani et al., 2003*). We crossed homozygous *Atoh1*^Flox/Flox^ females to males that were doubly heterozygous for *HoxA2::Cre*^TG^ and *Atoh1*^LacZ^ to obtain *Atoh1*^Flox/LacZ^;*HoxA2::Cre*^TG/+^ mice, hereafter called *Atoh1::HoxA2*-CKO mice. In situ hybridization for *Atoh1* RNA confirmed that conditional deletion only from the ITR (*Figure 6—figure supplement 1A*). ITR neurons were still present but mislocalized on the lateral side of the trigeminal motor nucleus instead of migrating medially at E14.5 (*Figure 6—figure supplement 1B and C*).

Because *Atoh1::HoxA2*-CKO mice survived the neonatal period (*Figure 6—figure supplement 2*), we could test their breathing behavior using unrestrained whole-body plethysmography (UWBP) at three weeks of age. We found that *Atoh1::HoxA2*-CKO mice had more sigh-induced and spontaneous apneas, without a change in the number of sighs or inter breath interval irregularity (*Figure 6Ai–Aiv*). Interestingly, *Atoh1::HoxA2*-CKO mice did have a smaller tidal volume per breath, resulting in a smaller minute ventilation than control littermates (*Figure 6Avi and Aviii*), phenotypes that were observed neither in our *Atoh1::En1*-CKO mice nor in *Atoh1::Phox2b*-CKO mice. Nevertheless, *Atoh1::HoxA2*-CKO mice did not have abnormal respiratory chemoresponses (*Figure 6B*). *Atoh1* is thus essential for normal ITR development, but loss of *Atoh1* from the ITR does not impair neonatal survival or respiratory chemoresponses. The abnormal chemoresponses caused by loss of *Atoh1* from the *En1-* or *Phox2b*-domain are thus caused by loss of rRL or abnormal development of RTN neurons, respectively.

### Loss of *Atoh1* from neurons involved in respiratory chemoresponses results in neonatal lethality

Through studying a series of conditional knockout mice, our lab has now mapped the role of all *Atoh1*-lineage neurons in neonatal survival (*Figure 7*). There is no single *Atoh1*-derived population that fully accounts for the perinatal death of *Atoh1*-null mice. In light of the fact that *Atoh1* loss from either the *En1* or *Phox2b* domains leads to abnormal chemoresponses, and that loss of *Atoh1* from the *Phox2b* domain results in neonatal lethality in only about half of the mice, we hypothesized that combined loss of *Atoh1*-derived neurons involved in respiratory chemoresponsiveness might be responsible for the 100% neonatal lethality of *Atoh1*-nulls.

To test this hypothesis, we crossed *Atoh1*^Flox/+^;*En1*^Cre/+^ males with *Atoh1*^LacZ/+^;*Phox2b::Cre*^TG/+^ females to obtain mice that lack *Atoh1* from both the *En1-* and *Phox2b*-intersectional domains (*Atoh1*^Flox/LacZ^;*En1*^Cre/+^;*Phox2b::Cre*^TG/+^ mice, hereafter *Atoh1*-dCKO mice). Out of nearly four hundred offspring observed, no *Atoh1*-dCKO mice survived past P1 (*Figure 7—figure supplement 1*). We did not find any alive *Atoh1*-dCKO mice, but several were found dead in the cage on the day of birth. These animals had turned blue and there was no visible milk spot, suggesting that they died of respiratory failure before drinking any milk. These results show that loss of *Atoh1* from chemoresponsive nuclei accounts for the 100% neonatal death seen in *Atoh1*-null mice (*Ben-Arie et al., 1997*).

## Discussion

Many of the transcription factors expressed in the developing hindbrain are necessary for respiratory circuit development and survival (*Gray, 2008*). In some cases, loss of a particular factor causes respiratory failure that is traceable to a single respiratory nucleus: for example, neonatal death in *Dbx1*-null mice is caused by loss of rhythmogenic preBötC neurons (*Bouvier et al., 2010*; *Wu et al., 2017*). The broad expression domains of factors such as *Atoh1*, *Tlx3* and *Lbx1*, however, have made it difficult to pinpoint their role in a specific nucleus or functional impairment (*Gray, 2008*; *Huang et al., 2012*; *Pagliardini et al., 2008*; *Shirasawa et al., 2000*). Here we used intersectional

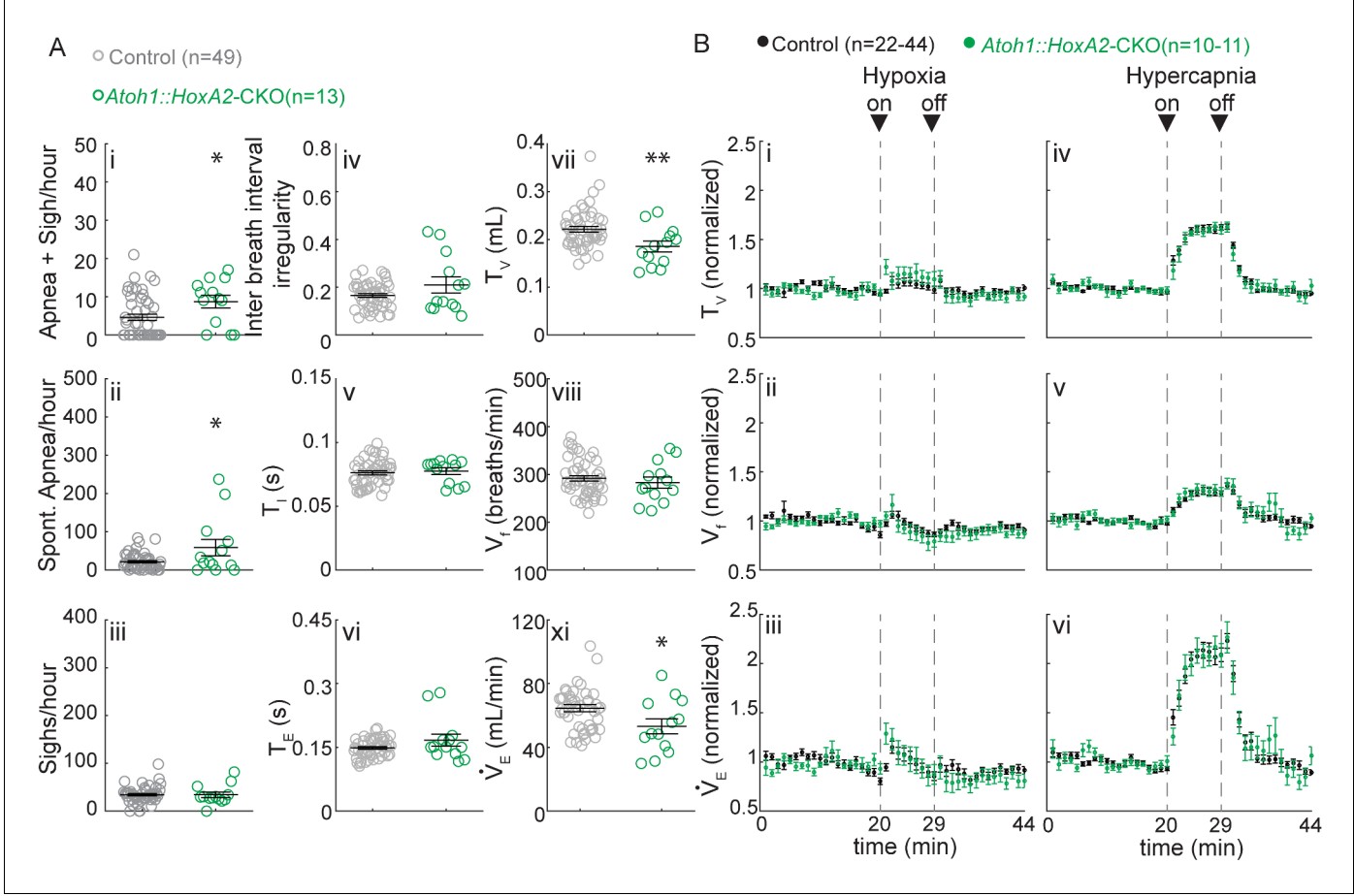

**Figure 6.** *Atoh1::HoxA2*-CKO mice have normal chemoresponses. (**A**) *Atoh1::HoxA2*-CKO mice have more apneas following sighs (i) and spontaneous apneas (ii) than control littermates and a smaller tidal volume per breath ($T_V$) (vi) resulting in smaller minute ventilation ($V_E$) (Viii). Other breathing parameters were not affected. Significance for room air breathing parameters were determined using a t-test (2-tailed). *p<0.05, **p<0.01. (**B**) *Atoh1:: HoxA2*-CKO mice have normal respiratory chemoresponses in hypoxia (i to iii) and hypercapnia (iv to vi). Significance was determined using a t-test (2-tailed) at each individual time point, *p<0.0011 (0.05/44 for Bonferroni correction). Error bars represent mean ± SEM.

DOI: https://doi.org/10.7554/eLife.38455.017

The following source data and figure supplements are available for figure 6:

**Source data 1.** Raw numbers observed surviving offspring.

DOI: https://doi.org/10.7554/eLife.38455.020

**Figure supplement 1.** *Atoh1::HoxA2*-CKO mice have abnormal migration of ITR neurons.

DOI: https://doi.org/10.7554/eLife.38455.018

**Figure supplement 2.** *Atoh1::HoxA2*-CKO mice survive the neonatal period.

DOI: https://doi.org/10.7554/eLife.38455.019

genetics to uncover a role for *Atoh1* in the development of chemoresponsive neuronal populations that also express *Engrailed1* and *Phox2b* (the PBC and RTN, respectively). Concomitant loss of *Atoh1* from these two domains causes fully penetrant neonatal lethality. This study thus answers the decades-old question about the cause of respiratory failure in *Atoh1*-null mice: they die of an inability to modulate respiratory rhythms in response to hypoxic and hypercapnic conditions.

Several pieces of evidence suggest that the abnormal chemoresponses we observed in the *Atoh1::En1*-CKO mice are primarily due to loss of *Atoh1*-lineage parabrachial neurons. First, even though multiple nuclei develop from the intersectional domain of *Atoh1* and *En1*, the parabrachial neurons were the only *Atoh1*-lineage neurons that were activated specifically during hypercapnic and hypoxic chemochallenges (*Figure 1*). Second, silencing the output from the cerebellar cortex does not cause abnormal respiratory chemoresponses (*Figure 3—figure supplement 1*). Third,

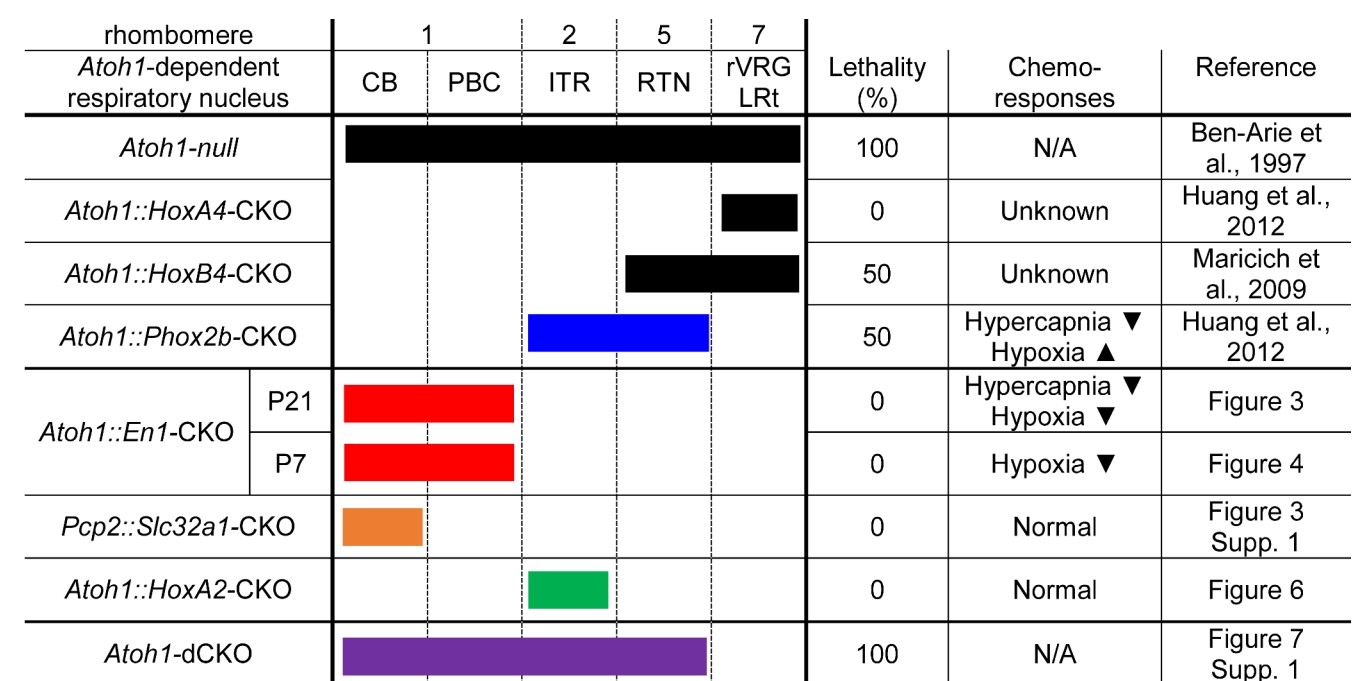

| rhombomere | 1 | | 2 | 5 | 7 | Lethality (%) | Chemo-responses | Reference |
|---|---|---|---|---|---|---|---|---|
| *Atoh1*-dependent respiratory nucleus | CB | PBC | ITR | RTN | rVRG LRt | | | |
| *Atoh1-null* | ██ | ██ | ██ | ██ | ██ | 100 | N/A | Ben-Arie et al., 1997 |
| *Atoh1::HoxA4*-CKO | | | | | ██ | 0 | Unknown | Huang et al., 2012 |
| *Atoh1::HoxB4*-CKO | | | | ██ | ██ | 50 | Unknown | Maricich et al., 2009 |
| *Atoh1::Phox2b*-CKO | | | ██ | ██ | | 50 | Hypercapnia ▼ Hypoxia ▲ | Huang et al., 2012 |
| *Atoh1::En1*-CKO P21 | ██ | ██ | | | | 0 | Hypercapnia ▼ Hypoxia ▼ | Figure 3 |
| *Atoh1::En1*-CKO P7 | ██ | ██ | | | | 0 | Hypoxia ▼ | Figure 4 |
| *Pcp2::Slc32a1*-CKO | ██ | | | | | 0 | Normal | Figure 3 Supp. 1 |
| *Atoh1::HoxA2*-CKO | | | ██ | | | 0 | Normal | Figure 6 |
| *Atoh1-dCKO* | ██ | ██ | ██ | ██ | | 100 | N/A | Figure 7 Supp. 1 |

Abbreviations: CB, Cerebellum; PBC, parabrachial complex; ITR, intertrigeminal region; RTN, retrotrapezoid nucleus; rVRG, rostral ventral respiratory group; LRt, lateral reticular. ▼ = impaired response. ▲ = enhanced response.

**Figure 7.** *Atoh1*-dependent neurons involved in chemoresponses and rates of neonatal lethality.
DOI: https://doi.org/10.7554/eLife.38455.021
The following figure supplement is available for figure 7:

**Figure supplement 1.** *Atoh1*-dCKO mice die in the neonatal period.
DOI: https://doi.org/10.7554/eLife.38455.022

differences in abnormal chemoresponses between *Atoh1::En1*-CKO mice and control littermates are observed long before the cerebellum fully matures or *Atoh1*-lineage cerebellar granule cells even make their first functional synapses *(Figure 4)*. Lastly, our results confirm and extend previous work showing that lesioning the parabrachial nucleus impairs respiratory chemoresponses (*Mizusawa et al., 1995*; *Song and Poon, 2009a*, *2009b*) and that the parabrachial nucleus projects to the ventrolateral medulla (*Fulwiler and Saper, 1984*; *Holstege, 1988*; *Yokota et al., 2015*). We found rostral *Atoh1*-lineage neurons projecting to the retrotrapezoid nucleus and preBötC, but we did not observe any projections to motor nuclei or specifically the reported connections with the hypoglossal nucleus. This discrepancy can be because we used genetic tools (*Atoh1*-driven) that only target a subset of PBC neurons. Furthermore, these genetic tools are less prone to potential off target effects caused by needle tracks and local injections. Therefore, we propose that *Atoh1*-lineage PBC neurons are an essential component in the chemosensory pathways that relays information from peripheral, and perhaps central, chemosensors to enhance the firing of central chemosensitive and rhythmogenic neurons within the respiratory network.

Nevertheless, we cannot exclude the possibility that *Atoh1*-lineage, deep cerebellar nuclei contribute to the observed phenotypes. *Atoh1*-lineage parabrachial and deep cerebellar neurons are born around the same embryonic day (E9.5-E12.5) (*Rose et al., 2009a*) and since we do not know what factors determine their differentiation, there are no developmental or genetic tools to target the deep cerebellar nuclei without affecting the parabrachial nucleus. Viral approaches do not work, either, as we discovered: deep cerebellar neurons send collateral projections throughout the

cerebellum and the brainstem, so viral injections infect *Atoh1*-lineage deep cerebellar neurons regardless of the injection site. Nevertheless, the primary source of input to the deep cerebellar nuclei are the Purkinje cells, and silencing these cells did not recapitulate or alter respiratory chemoresponses. This latter finding, together with our data showing absence of chemoresponses in deep cerebellar neurons, make them unlikely to be essential for respiratory chemoresponses.

Likewise, *Atoh1* loss from the intertrigeminal region had little effect on respiratory chemoresponses. In agreement with earlier studies from our lab, we found that intertrigeminal neurons depend on *Atoh1* expression for normal migration, which resulted in increased number of apneas and a shallower breaths. Previous studies suggested a role for intertrigeminal neurons in the attenuation of sleep and reflex apneas (*Radulovacki et al., 2003*, *2004*). We indeed saw an increase in sigh-induced apneas but we saw an increase in the number of spontaneous apneas in only two out of thirteen *Atoh1::HoxA2*-CKO mice, suggesting that the previously reported apneas were not likely caused by a central mechanism (*Figure 6*). Despite the changes in respiratory control in room air, we did not observe any abnormal respiratory chemoresponses in *Atoh1::HoxA2*-CKO mice. These results support the hypothesis that the respiratory phenotype of mice that lack *Atoh1* in both ITR and RTN (the *Atoh1::Phox2b*-CKO mice) results from abnormal RTN development.

The current work provides evidence that there are two *Atoh1*-lineage nuclei that rely on *Atoh1* expression for their function in respiratory chemoreflexes: the PBC and RTN. Removal of *Atoh1* from both of these nuclei recapitulated the fully penetrant neonatal lethality of *Atoh1*-null mice, which we hypothesize is caused by combined loss of $CO_2$-evoked glutamatergic signaling (RTN) and hypoxia-induced activation of PBC neurons. Previous studies in neonatal models have shown that arterial $CO_2$ unloading removes respiratory drive and results in sustained apnea, in accordance with the notion that $CO_2$ sensing is essential for neonatal breathing (*Nattie, 1999*; *Praud et al., 1997*). We suspect that, when directly or indirectly activated during respiratory chemochallenges, *Atoh1*-lineage neurons increase glutamatergic input to the preBötzinger complex, which is thought to form the central respiratory pattern generator. Since neither *Atoh1::En1*-CKO mice nor *Atoh1::Phox2b*-CKO mice show complete lethality nor complete loss of respiratory chemoresponses, it is likely that these two pathways of chemosensation (PBC and RTN-mediated) are partially redundant, as would make sense for any circuit that controls a process as fundamental to survival as breathing. Indeed, en bloc recordings from *Atoh1*-null mice reveal that slow respiratory frequency can be fully restored by application of a glutamate reuptake inhibitor and partially restored by substance P application (*Rose et al., 2009b*). Our results are also in agreement with previous recordings in wild-type brainstems, where loss of input from the pons and RTN to the ventrolateral medulla results in a transition from a tri-phasic to a slow gasp-like, mono-phasic rhythm (*Rubin et al., 2009*; *Smith et al., 2009*). The present study provides further evidence that pontine PBC and RTN neurons are essential for the respiratory versatility observed in vivo. Despite the normal development of preBötzinger neurons that are sufficient for in situ respiratory rhythms, excitatory projections from *Atoh1*-lineage neurons are necessary for normal breathing in vivo.

Caffeine treatment, along with glutamate and substance P, can stimulate endogenous, slow respiratory rhythms in rats (*Ruangkittisakul et al., 2010*); this is thought to be the mechanism by which caffeine reduces apneas of prematurity in newborn infants (*Aranda et al., 1977*; *Natarajan et al., 2007*). Caffeine drives this excitation mainly by acting as an antagonist to the A1 adenosine receptor that inhibits respiratory frequency (*Koos et al., 2001*). Decreased inhibition might compensate for the loss of *Atoh1*-dependent glutamatergic drive in *Atoh1::En1*-CKO mice, thereby rescuing the apnea. Caffeine treatment did not, however, improve the respiratory chemoresponses in *Atoh1::En1*-CKO mice, suggesting that increased ventilation during chemochallenge relies on more precise signaling of distinct neurons in the circuitry. This is in agreement with the observation that some, but not all, *Atoh1*-dependent parabrachial neurons are activated during chemochallenges (*Figure 1*). To our surprise, caffeine treatment actually decreased respiratory output in control animals, which might be caused by inhibition of A2 adenosine receptors that increase respiratory frequency (*Koos, 2011*; *Koos and Chau, 1998*; *Koos et al., 2001*). Future studies with specific adenosine agonists and antagonists are needed to elucidate the specific effects of long-term caffeine treatment on respiratory control in neonatal mice.

The breathing abnormalities of *Atoh1::En1*-CKO mice resemble those seen in premature infants: increased apneas and sighs, abnormal rhythms, attenuated responses to hypercapnia, and, perhaps most surprisingly, respiratory suppression in hypoxia (*Martin et al., 2004*; *Abu-Shaweesh, 2004*).

These also mimic respiratory responses to hypoxia that occur in utero, when the oxygenation of fetal blood is regulated by the mother. During development, exposure to low $O_2$ suppresses fetal movements, including breathing movements, likely to limit oxygen demand in relatively anaerobic environments. Several reports have suggested that pontine nuclei play a role in this respiratory suppression (*Gluckman and Johnston, 1987*; *Haddad and Mellins, 1984*) and in the postnatal maturation of hypoxic responses (*Bissonnette and Knopp, 2001*; *Waters and Gozal, 2003*). Nevertheless, it was not known precisely which neurons were important for adaptation to hypoxia. Given that abnormal development of pontine nuclei and mutations in *En1* have been observed in several sudden infant death syndrome (SIDS) cases (*Lavezzi, 2015*, *2016*, *2004*; *Weese-Mayer et al., 2004*), we hypothesize that hypoxia-mediated respiratory suppression concomitant with immature responses to hypercapnia might be a major risk factor for SIDS.

# Materials and methods

## Key resources table

| Reagent type (species) or resource | Designation | Source or reference | Identifiers |
|---|---|---|---|
| Strain: C57BL/6 | Rosa$^{lsl-tdTomato}$ | The Jackson Laboratory | RRID: IMSR_JAX:007914 |
| Strain: C57BL/6 | Atoh1$^{Cre}$ | *Yang et al., 2001* | RRID: MGI:4844110 |
| Strain: C57BL/6 | En1$^{Cre}$ | The Jackson Laboratory | RRID: IMSR_JAX:007916 |
| Strain: C57BL/6 | Rosa$^{lsl-LacZ}$ | The Jackson Laboratory | RRID:IMSR_JAX:012429 |
| Strain: C57BL/6 | Atoh1$^{LacZ}$ | The Jackson Laboratory | RRID:IMSR_JAX:005970 |
| Strain: C57BL/6 | Atoh1$^{Flox}$ | The Jackson Laboratory | RRID:MGI:4420944 |
| Strain: C57BL/6 | Slc32a1$^{Flox}$ | The Jackson Laboratory | RRID:IMSR_JAX:012897 |
| Strain: C57BL/6 | Pcp2::Cre$^{TG}$ | The Jackson Laboratory | RRID:IMSR_JAX:004146 |
| Strain: C57BL/6 | HoxA2::Cre$^{TG}$ | *Awatramani et al. (2003)* | N/A |
| Strain: C57BL/6 | Phox2b::Cre$^{TG}$ | The Jackson Laboratory | RRID: IMSR_JAX:016223 |
| Strain: C57BL/6 | Atoh1$^{FlpO}$ | This paper: *Figure 5—figure supplement 1* | N/A |
| Strain: C57BL/6 | Rosa$^{FSF-LSL-tdTomato}$ | The Jackson Laboratory | RRID:IMSR_JAX:021875 |
| Strain: C57BL/6 | Sox2::Cre$^{TG}$ | The Jackson Laboratory | RRID:MGI:3801167 |
| Antibody | anti-cFos (rabbit polyclonal) | Santa Cruz | SC-52; RRID:AB_2106783 |
| Antibody | anti-NK1R (rabbit polyclonal) | Advanced Targeting Systems | AB-N04; RRID: AB_171801 |
| Antibody | anti-CGRP (rabbit polyclonal) | Sigma Aldrich | C8198; RRID:AB_259091 |
| Antibody | anti-PACAP (mouse monoclonal) | Abcam | ab216589 |
| Antibody | anti-synapsin (mouse monoclonal) | Synaptic Systems | 106 001; RRID:AB_2617071 |
| Antibody | anti-TH (rabbit polyclonal) | ImmunoStar | 22941; RRID:AB_572268 |
| Antibody | anti-ChAT (goat polyclonal) | EMD millipore | AB144P; RRID:AB_2079751 |
| Chemical compound, drug | Caffeine | FISHER | S25215A |
| Chemical compound, drug | X-gal | Gold Biotechnology | X4281C |
| Commercial assay or kit | Caffeine/Pentoxifylline ELISA | Neogen | 106419 |
| Software, algorithm | MATLAB | MathWorks | RRID: SCR_001622 |
| Software, algorithm | Ponemah 3 | DSI | N/A |

*Continued on next page*

*Continued*

| Reagent type (species) or resource | Designation | Source or reference | Identifiers |
|---|---|---|---|
| Software, algorithm | FinePoint | DSI | N/A |
| Software, algorithm | ImageJ | NIH | RRID: SCR_003070 |

## Mouse lines

All animals were housed in a Level 3, AALAS-certified facility on a 14 hr light cycle. Husbandry, housing, euthanasia, and experimental guidelines were reviewed and approved by the Institutional Animal Care and Use Committee (IACUC) of Baylor College of Medicine. The following genetically engineered mouse lines were used: $Rosa^{lsl-tdTomato}$ ($Gt(ROSA)26Sor^{tm9(CAG-tdTomato)Hze}$, JAX:007914), $Atoh1^{Cre}$ (**Yang et al., 2001**), $En1^{Cre}$ ($En1^{tm2(cre)Wrst/J}$, JAX:007916):, $Rosa^{lsl-LacZ}$ ($Gt(ROSA)26Sor^{tm1(CAG-lacZ,-EGFP)Glh}$, JAX:012429), $Atoh1^{LacZ}$($Atoh1^{tm2Hzo}$, JAX:005970), $Atoh1^{Flox}$($Atoh1^{tm3Hzo}$, MGI:4420944), $Pcp2^{Cre}$ (B6.129-Tg(Pcp2-cre)2Mpin/J, JAX: 004146), $Slc32a1^{Flox}$ (Slc32a1tm1Lowl/J, JAX: 012897), $HoxA2::Cre^{TG}$ (**Awatramani et al., 2003**), $Phox2b::Cre^{TG}$ (Tg(Phox2b-cre)3Jke, JAX:016223), $Atoh1^{FlpO}$, $Ai65$ ($Gt(ROSA)26Sor^{tm65.1(CAG-tdTomato)Hze}$,JAX:021875), and $Sox2::Cre^{TG}$ (Tg(Sox2-cre)1Amc, MGI:3801167). Ear tissue, collected from ear-marking, was used for PCR genotyping to minimalize stress on animals. For timed pregnancies, noon on the day of the vaginal plugging was set as embryonic day 0.5 (E0.5).

## Generation of the $Atoh1^{FlpO}$ Mice

The *FlpO* sequence from *pQUAST-FLPo* was ligated to a PGK-Neo cassette flanked by lox2722 sites in a pUC vector. This *FlpO-PGK-Neo* cassette was then cloned into a pre-existing pBlueScript II KS + plasmid that contained the *Atoh1* 5' and 3' targeting arms without disrupting the *Atoh1* transcriptional start site, identical to the approach previously described (**Rose et al., 2009b**). This construct was then electroporated into B57/6J ES cells with an agouti mutation. These ES cells were expanded under neomycin selection and screened for correct recombination by Southern blot using external 3' probes, and internal 5' PCR. Six clones were further expanded, tested correct recombination using PCR and sequence validated. Three clones were chosen to be injected into albino B57/6J blastocyst to generate chimeras. Chimeras were backcrossed to albino B57/6J females to generate heterozygote $Atoh1^{FlpO-Neo}$ mice. These were crossed to $Sox2^{Cre}$ to remove the *PGK-Neo* cassette. See ***Figure 5—figure supplement 1*** for genomic targeting diagram, and example PCR genotyping (forward primer: CTTCGTTGCACGCGAC, reverse primer: CACAATTTATCGTGTAGCCG. WT: 2.2 kb, FlpO-KI: 2.6 kb).

## Immunofluorescence (IF) Assays

IF and cryosectioning were performed using previously described protocols (**Huang et al., 2012**). Mice older than one week were perfused with 4% PFA, after which brains were dissected and post-fixed overnight in 4% PFA at 4°C. Embryonic and P0 brains were directly dissected from pups and dropped fixed in 4% PFA at 4°C. After fixation brains were cryopreserved in 30% sucrose solution in PBS until sunk and frozen in OCT. Frozen sections were cut at 40 µm and kept at 4°C in the dark until used for immunostaining. For immunolabeling sections were first blocked in 5% normal donkey serum, 0.5% Triton-X in PBS for one hour at room temperature. Next, sections were incubated in primary antibody overnight at 4°C in blocking solution, followed by three washes and secondary labeling for at least two hours at room temperature. Nuclei were labeled using DAPI (1:10,000) and slides were mounted in DAPI-free mounting solution (Vectashield). The following antibodies and their dilutions were used: rabbit anti-cFos (1:5000, Santa Cruz), rabbit anti-NK1R (1:2000, Advanced Targeting Systems), mouse anti-Synapsin (1:1000, Synaptic Systems), goat anti-TPH2 (1:1000, Santa Cruz), mouse anti-TH (1:200, ImmunoStar), and goat anti-ChAT (1:200, EMD Millipore). Secondary antibodies were conjugated with Alexa Fluor 488 (1:1000, Molecular Probes). We used a Leica TCS SP5 confocal microscope system to image fluorescent signal. Image brightness and contrast were normalized using ImageJ.

## X-gal staining

Embryos were collected at E14.5, brains were dissected and fixed for 10 min in ice-cold formalin followed by brief washing in PBS on ice and in room-temperature wash buffer. β-galactosidase activity was assayed by embedding and sectioning tissue as previously described (*Ben-Arie et al., 2000*). X-gal stained whole brains were stored at 70% ethanol. X-gal stained sections were first counterstained with nuclear fast red (Vector laboratories) and then imaged using a bright-field Axio Imager M2 microscope, equipped with an Axio Cam MRc5 color camera (Carl Zeiss, Germany). Contrast and saturation were adjusted using ImageJ and Adobe Photoshop.

In situ hybridization (ISH) was performed on 25μm-thick sagittal sections cut from unfixed, fresh frozen E14.5 embryonic control and CKO mice covering the entire embryo. We generated digoxigenin (DIG)-labeled mRNA antisense probes against *Atoh1* using reverse-transcribed mouse cDNA as a template and a RNA DIG-labeling kit from Roche. Primer and probe sequences for the probe is available on the Allen Brain Atlas website (www.brain-map.org) and were validated previously (*Huang et al., 2012*). ISH was performed by the RNA In Situ Hybridization Core at Baylor College of Medicine using an automated robotic platform as previously described (*Yaylaoglu et al., 2005*).

## Assessing cFos expression

To analyze in vivo hypoxia or hypercapnia induced cFos expression, adult animals (6–8 weeks old) were habituated to the plethysmography chambers five hours the day prior to the experiment and one hour on the day of the experiment. Freely moving mice were placed in whole-body plethysmography chambers (Buxco) through which fresh air was pumped at a basal flow rate of 0.5 L/min. Next, animals were exposed to either room air, hypoxia (10% $O_2$, balance $N_2$), or hypercapnia (5% $CO_2$, 21% $O_2$, balance $N_2$) for 1 hr. Animals were sacrificed within 30 min of exposure and tissue was processed as described in 'Immunofluorescence (IF) Assays.'

## P21 juvenile unrestrained whole-body plethysmography (UWBP)

Breathing analysis was performed using previously reported protocols with minor modifications (*Huang et al., 2012*; *Orengo et al., 2018*; *Yeh et al., 2017*). To test respiratory parameters, weaning age mice (P19-21) were placed in the plethysmography chambers and habituated for at least one hour prior to the experiment. After this habituation period, we recorded twenty minutes of room air breathing to determine the baseline. Next, we exposed animals to nine minutes in either hypoxia (10% $O_2$, balance $N_2$) or hypercapnia (5% $CO_2$, 21% $O_2$, balance $N_2$). After this chemochallenge, we recorded breathing behavior for another fifteen-minute recovery period in room air. To determine changes in breathing behavior in response to the chemochallenge, we normalized all breathing parameters to baseline values using the following formula: normalized $y(t)=y(x)/(average(y(x_{baseline}))$, where y is the average value of a parameter during any given minute (x) and baseline is minutes one to twenty.

We used Phonemah three software (DSI) to identify breath waveforms and used custom-written MATLAB (Mathworks) code to derive inspiratory time, expiratory time, and tidal volume (*Source code 1*). Breaths with an inspiration time shorter than 0.03 s or an expiration time longer than 10 s were excluded. Minute long segments in which the average breathing frequency (breaths/min) was higher than 500 were excluded from our analysis to prevent confounds from sniffing or exploring. Tidal volume was adjusted for body temperature as previously described (*Ray et al., 2011*). Apneas were defined as breaths longer than 0.5 s and at least twice the length of the average of the six surrounding breaths (three previous and three following). Apneas were divided in spontaneous apneas and apneas that were preceded by a sigh (or a sigh in the previous breath). Sighs were defined as breaths with an inspiratory tidal volume 2.5 as big and an inspiratory time 1.25 as long as the previous five breaths. Inter Breath Interval (IBI) irregularity was defined as: IBI irregularity = abs (breath length(n + 1)-breath length(n))/breath length(n). Minute ventilation was defined as the total amount of air breathed per minute (breathing frequency * tidal volume). Source data from the plethysmography analysis can be found in the source data files: *Figure 2—source data 1*, *Figure 3—source data 1*, *Figure 3—figure supplement 1—source data 1*, and *Figure 6—source data 1*.

Since we did not observe any difference in breathing parameters between male and female mice, or between *Atoh1*-heterozygote animals and any of the Cre-lines, we grouped these animals together in the analysis.

## P7 neonatal unrestrained whole-body plethysmography (UWBP)

Breathing analysis for neonatal mice was performed using a DSI pup-plethysmography set-up designed especially for these experiments. Specialized, small pup plethysmography chambers (DSI) were connected to a FinePointe Whole Body Plethysmography Unit with gas switch capability (DSI). The unit provided the pup plethysmograph chambers with a constant 1 L/min airflow and amplified breathing waveforms. P7 (day of birth is P0) animals were placed in the plethysmography chambers and could acclimate for 10 min, followed by 5 min of room air recording and a 5 min gas-challenge (hypoxia (10% $O_2$, balance $N_2$) or hypercapnia (5% $CO_2$, 21% $O_2$, balance $N_2$)).

FinePointe software was used to define breaths and calculate basic breathing parameters. Breaths were defined as followed: (1) peak inspiratory flow (PIF) must be before peak expiratory flow (PEF); (2) expiration or inspiration is less than a couple samples; (3) Rpef, EF50, and Tr can be computed; (4) conditioning coefficient can be computed; (5) MinimumBoxFlowThreshold is larger than 0.005; (6) Start of expiration can be identified on the AC flow waveform. Breaths were rejected if $T_V$ was smaller than 0.005 ml or larger than 1.0 ml; if $T_I$ was larger than 0.02 s or twice as large as $T_E$; or if inspiratory volume was not within 50% to 150% of the couple expiratory volume. For each minute of breathing recording a rejection index (RINX) was calculated as the percentage of the trace from which individual breaths were defined (length breathing trace used for analysis/one minute * 100%). For our final analysis, we included only those minutes for which the RINX was less than 40%. All animals from which we recorded fewer than three reliable minutes of respiratory rhythms in both room air and during chemochallenge (RINX <40%) were excluded from our analysis. The percentage of animals of each genotype included in the final analysis were similar (23 control mice out of 33 (70%); 12 *Atoh1::En1*-CKO mice out of 16 (75%); Chi-Square = 0.133, p=0.7149). RINX of included animals was also similar between genotypes (control mice: 18%; *Atoh1::En1*-CKO mice: 19.8%; t-test, p=0.614). From the included minutes the following parameters were derived: inhalation time, exhalation time, breathing frequency, tidal volume, and minute ventilation. Apneas and sighs were determined by a trained observer who was blinded to the genotype of the mice.

Source data from the pup plethysmography analysis can be found in the source data file: *Figure 4—source data 1*.

## Caffeine administration and ELISA

Caffeine was administered to pups through the drinking water of lactating dams (0.3 gram/L) from postnatal day three until weaning. Caffeine was dissolved in water provided by the animal facility and stored in special dark bottles to prevent photodegradation. Plasma levels of caffeine in pups were determined using a commercially available Caffeine/Pentoxifylline ELISA kit (Neogen). We obtained blood and isolated plasma from P19 pups raised in cages with normal water or caffeinated water.

## Quantification and statistical analysis

All quantification and statistical analyses were performed using MATLAB (Mathworks). Data in text and figures represent the mean ± standard error of the mean (SEM). Student's t-test was used when comparing two independent groups. Chi-square test were performed to test Mendelian ratios of surviving offspring. We used two-way ANOVA to determine interaction effects between genotype (control vs. conditional knockout) and treatment (no caffeine vs. caffeine), followed by a Tukey-Kramer post-hoc analysis in case significance was reached. To determine time-specific significant differences in respiratory chemoresponses, serial t-tests or ANOVA's were performed and the *p*-value was Bonferroni adjusted for multiple comparisons. Statistical significance was accepted at p<0.05 for all other tests.

## Acknowledgements

We thank Dr. T Klisch for expertise and guidance generating the *Atoh1*[FlpO] knock-in mouse line, Dr. R Ray for continuous advice on project design, and V Brandt for her insightful comments and edits

on the manuscript. We also thank Dr. S Dymecki for editorial and scientific comments to the manuscript and for sharing the *HoxA2::Cre*[TG] mice. We thank Drs. J Elmquist, L Gan, A Joyner, and R Sillitoe for sharing the *Phox2b::Cre*[TG], *Atoh1*[Cre], *En1*[Cre] mice, and *Pcp2*[Cre/+];*Slc32a1*[Flox/Flox] respectively. This project was supported by the Neurobehavioral Core facility (U54HD083092), the Neuroconnectivity Core facility (U54HD083092), and the RNA In Situ Hybridization Core facility (with the expert assistance of Cecilia Ljungberg, Ph.D., 1S10 OD016167 and IDDRC grant 1U54 HD083092, Eunice Kennedy Shriver National Institute Of Child Health and Human Development). This work was supported by American Heart Association AWRP Predoctoral Fellowship to MEH (17PRE33660616). HYZ is a Howard Hughes Medical Institute investigator.

## Additional information

### Competing interests

Huda Y Zoghbi: Senior editor, *eLife*. The other author declares that no competing interests exist.

### Funding

| Funder | Grant reference number | Author |
| --- | --- | --- |
| American Heart Association | Predoctoral fellowship award number 17PRE33660616 | Meike E van der Heijden |
| Howard Hughes Medical Institute | | Huda Y Zoghbi |

The funders had no role in study design, data collection and interpretation, or the decision to submit the work for publication.

### Author contributions

Meike E van der Heijden, Conceptualization, Data curation, Formal analysis, Funding acquisition, Validation, Investigation, Visualization, Methodology, Writing—original draft, Writing—review and editing; Huda Y Zoghbi, Conceptualization, Supervision, Funding acquisition, Writing—review and editing

### Author ORCIDs

Meike E van der Heijden (iD) http://orcid.org/0000-0003-0801-8806
Huda Y Zoghbi (iD) http://orcid.org/0000-0002-0700-3349

### Ethics

Animal experimentation: All animals were housed in a Level 3, AALAS-certified facility on a 14hr light cycle. Husbandry, housing, euthanasia, and experimental guidelines were reviewed and approved by the Institutional Animal Care and Use Committee (IACUC) of Baylor College of Medicine (protocol number: AN1013).

### Decision letter and Author response

Decision letter https://doi.org/10.7554/eLife.38455.027
Author response https://doi.org/10.7554/eLife.38455.028

## Additional files

### Supplementary files

• Source code 1. Custom MATLAB code.
DOI: https://doi.org/10.7554/eLife.38455.023

• Transparent reporting form
DOI: https://doi.org/10.7554/eLife.38455.024

## Data availability

All data generated or analysed during this study are included in the manuscript and supporting files. Source data files have been provided for Figures 2, 3, 4 and 6.

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
