## [Decision Letter]

Thank you for submitting your article "Loss of *Atoh1* from neurons regulating hypoxic and hypercapnic chemoresponses causes neonatal respiratory failure in mice" for consideration by *eLife*. Your article has been reviewed by two peer reviewers, including Jan-Marino Ramirez as the Reviewing Editor and Reviewer #1, and the evaluation has been overseen Eve Marder as the Senior Editor. The following individual involved in review of your submission has agreed to reveal their identity: Daniel K Mulkey (Reviewer #2).

The reviewers have discussed the reviews with one another and the Reviewing Editor has drafted this decision to help you prepare a revised submission.

Summary:

This manuscript by Meike van der Heijden and Huda Zoghbi significantly contributes to our understanding of the role of the parabrachial complex. The parabrachial complex (PBC) is a sensory integration center at the interface between forebrain structures and the brainstem that has long been implicated in respiratory control including serving as a chemotransduction relay station. But, the problem is that this area is very heterogeneous, and most what we know about this area is based on somewhat outdated studies involving often crude lesioning and non-specific electrical or chemical stimulations. This study by van der Heijden and Zoghbi overcomes these limitations by showing *Atoh1*-lineage neurons in the PBC contribute to central and peripheral chemoreflexes and together with *Atoh1* neurons in the retrotrapezoid nucleus are essential for survival. The reviewers were very impressed by this study: The experiments are well designed, controlled and the conclusions are generally well founded.

Essential revisions:

1) Apneas associated with sighs are mechanistically different from spontaneous apneic events. Indeed, the authors find in different mutants differential effects on the number of sighs versus apneas. Thus, we suggest that central apneas and apneas associated with sighs should be tabulated separately.

2) The authors should justify why they define apnea as breaths lasting > 0.5s, specially since the apnea on Figure 2A is not that different from baseline. It seems like apneas should be defined for each genotype based on inter-breath interval. While clinicians often use a certain interval without a breath, traditionally, apneas in mice are defined by the prolonged expiratory time relative to baseline (e.g. 4 times the eupneic expiratory time). We don't ask the authors to reanalyze their data, but this should be clarified and discussed.

3) Of interest are also the downward deflections in the plethysmograph recording (see e.g. Figures 2A, 4A): Were these expiratory efforts that are uncoupled with inspiration? This is just a comment and we don't ask the authors to analyze these downward deflections. But, we are curious to know whether expiration was abnormal in these mice.

4) The authors note that some genotypes show more irregular breathing compared to control. However, it is not clear to me whether this is a bad thing. For example, high heart rate variability is considered more stable, whereas less variability is associated with ensuing heart problems. If the same is true for respiratory control then further explanation is needed.

5) Baseline respiratory frequencies are relatively high compared to most mouse strains (PMID: 7977867). Also, the Materials and methods states that frequencies >500 are excluded; however, we would be concerned about anything >250 bpm, because the animals might have been stressed. Can the authors try some controls with extra-long acclimation periods, perhaps in an isolated room and cover the plethysmography chamber to prevent visual distractions? If frequencies are still high, then it would be more convincing that the frequencies that they observed were indeed baseline frequencies. The authors could also consider performing Poincaré analysis of respiratory variability as described (PMID: 26068853) to support the possibility that experiments were performed on quiescent mice.

6) Caffeine treatment normalized apneas and irregular breathing in *Atoh1^En1^*^-*CKO*^ mice but did is also improve life span? This would strengthen the link between irregular breathing and mortality.

7) The differential effects of caffeine on control and *Atoh1^En1-CKO^* suggest differential roles of adenosine signaling. That would be very interesting particularly since adenosine is thought to contribute to respiratory roll-off during hypoxia. In any case, caffeine has been shown to increase baseline breathing and chemoreception in several animals including humans (PMID: 18703176) so we agree with the authors that it is very surprising caffeine decreased respiratory output in control mice: This unusual response could be due to the high baseline frequency. There are certainly also other explanations: caffeine also inhibits A2 receptors, which are excitatory so perhaps this is an A2 dependent effect? The authors should discuss these possibilities. Of course, the authors could also try selective A1 and A2 blockers to test this possibility, but we feel that this would exceed the scope of the present study.

8) We would appreciate if the authors would also discuss whether they propose that PBC neurons are primary chemosensory neurons or a requisite component of the pathway.

9) Previous evidence showed that CO_2_/H^+^-sensitive PBC neurons are glutamatergic and express calcitonin gene-related peptide (CGRP) (PMID: 29103805; PMID: 25424719). While we don't ask for additional experiments, it would be helpful to know if these markers overlap with *Atoh1*.

10) A proposed role of CO_2_/H^+^-sensitive PBC neurons is maintenance of airway patency and there is evidence suggesting subsets of PBC neurons project to respiratory motor neurons (PMID: 25424719). However, results presented here (Figure 5) do not support this possibility. These differences should be discussed.

11) The 100% neonatal death in *Atoh1* null mice is very interesting. Did the authors observe whether the mice took a breath? Did they turn blue? Clarifying this would be very important for follow up studies.

---

## [Author Response]

Essential revisions:1) Apneas associated with sighs are mechanistically different from spontaneous apneic events. Indeed, the authors find in different mutants differential effects on the number of sighs versus apneas. Thus, we suggest that central apneas and apneas associated with sighs should be tabulated separately.

We thank the reviewers for this excellent suggestion. We have run our analysis again and this time we defined apneas are breaths lasting 0.5s and at least 2 times the length of the closest 6 breaths (3 previous breaths and 3 following breaths) (see also essential revision 2). This definition of apnea was used in many previous published papers (PMID: 28831138; PMID: 28392070; PMID: 27194336; PMID: 22653057). We defined apneas associated with sighs as apneas where that breath also met the criteria of a sigh or that were preceded by a sigh or if the prior breath was a sigh. Spontaneous apneas were those not associated with a sigh. We updated Figure 2, Figure 3—figure supplement 1, and Figure 6 accordingly.

2) The authors should justify why they define apnea as breaths lasting > 0.5s, specially since the apnea on Figure 2A is not that different from baseline. It seems like apneas should be defined for each genotype based on inter-breath interval. While clinicians often use a certain interval without a breath, traditionally, apneas in mice are defined by the prolonged expiratory time relative to baseline (e.g. 4 times the eupneic expiratory time). We don't ask the authors to reanalyze their data, but this should be clarified and discussed.See essential revision 1.3) Of interest are also the downward deflections in the plethysmograph recording (see e.g. Figures 2A, 4A): Were these expiratory efforts that are uncoupled with inspiration? This is just a comment and we don't ask the authors to analyze these downward deflections. But, we are curious to know whether expiration was abnormal in these mice.

We did not find that these downward deflections were overrepresented in either genotype. We did not find any evidence that expiration was abnormal in these mice.

4) The authors note that some genotypes show more irregular breathing compared to control. However, it is not clear to me whether this is a bad thing. For example, high heart rate variability is considered more stable, whereas less variability is associated with ensuing heart problems. If the same is true for respiratory control then further explanation is needed.

Irregular breathing is often observed in neonates (PMID: 15050210), patients with Cheynes-Stokes breathing (PMID: 4571351), Rett syndrome patients (PMID: 9452925), and a mouse model for congenital central hypoventilation syndrome (PMID: 18198276). High number of apneas can cause a higher interbreath interval irregularity, because the apnea is often at least twice as long as the breath preceding and following the apnea. However, irregularity scores are more sensitive than any kind of automated apnea counter, because it also takes in account breaths that just fail to make a cut-off for the apnea definition but are still remarkably longer than the prior breath. For examples of such breaths, see outlier points in Figure 1—figure supplement 2, particularly in the plot for Trace #340. Abnormalities in chemosensation can cause irregular breathing because of a delayed respiratory response to fluctuating O_2_ and CO_2_ levels (PMID: 4571351). The irregularity score is thus another measure of in vivo respiratory control that can point to abnormalities in central respiratory control.

5) Baseline respiratory frequencies are relatively high compared to most mouse strains (PMID: 7977867). Also, the Materials and methods states that frequencies >500 are excluded; however, we would be concerned about anything >250 bpm, because the animals might have been stressed. Can the authors try some controls with extra-long acclimation periods, perhaps in an isolated room and cover the plethysmography chamber to prevent visual distractions? If frequencies are still high, then it would be more convincing that the frequencies that they observed were indeed baseline frequencies. The authors could also consider performing Poincaré analysis of respiratory variability as described (PMID: 26068853) to support the possibility that experiments were performed on quiescent mice.

We performed Poincaré analysis for five randomly picked mice from each group. We found only highly variable breathing in the conditional knockout mice, likely due to the high occurrence of apneas, most other points fell along the diagonal line (Author response image 1). We also generated a cumulative frequency distribution curve, that shows untreated mice breathe at the same frequencies, whereas the CKO mice treated with caffeine breathe faster (left shifted curve) and control mice treated with caffeine breathe slower (right shifted curve). Slower breathing can be perceived as calmer or more acclimated mice. However, the CKO and control mice were always tested at the same time, in cohorts of at least 8 mice, yet the caffeine effect was in the opposing direction for these mice. These data, therefore, represent the same results as what was found in Figure 2— supplement 1, and we decided to show it to reviewers but not to incorporate it in the paper.

**Author response image 1. respfig1:** Caffeine had opposing effects on respiratory frequency distributions in *Atoh1^En1-CKO^*mice and control littermates. (A-D) Poincaré plots of breathing frequency of subsequent breaths plotted as the inter breath interval against subsequent breaths (n vs. n+1). Each plot represents the first 250 recorded breaths of five randomly selected animals from each group. Points that fall far away and a wider distribution from the dotted diagonal represent breaths that are more irregular. Breaths with an IBI larger than 0.5 seconds are possibly apnea. For each group, the plots were ordered from most irregular breathing to most regular breathing animals to ease visual comparison. (**E**) Cumulative frequency distribution curves *Atoh1^En1-CKO^*and control mice treated with or without caffeine. Untreated control and *Atoh1^En1-CKO^* mice had similar frequency distributions. The frequency distribution curve for control mice was right-shifted, representing longer IBIs and possibly calmer mice. The frequency distribution curve for *Atoh1^En1-CKO^* mice was left-shifted, representing shorter IBIs and potentially less calm mice. Significance was determined using a two-way repeated measure ANOVA. *p<.05. **p<.01. ***p<0.001. ****p<.0001. Error bars represent mean ± SEM.

We also performed experiments in mice with acclimation periods over 1.5 hours and did not observe a decline in respiratory frequencies. We have performed similar experiments in mature/adult mice and we found that these mice have a substantially smaller respiratory rate after the acclimation period used for this study (~200 breaths/minute). Furthermore, after acclimating mice for about 30 minutes they appeared quiescent and often asleep, suggesting that this relatively high baseline-breathing rate is a representation of age rather than stress.

6) Caffeine treatment normalized apneas and irregular breathing in Atoh1^En1-CKO^ mice but did is also improve life span? This would strengthen the link between irregular breathing and mortality.

Caffeine treatment did not normalize lifespan. Because cerebellar neurons are lacking in these mice, we believe the lifespan is reduced because of impaired coordination and compromised access to food and water. This is stated more clearly now in the text as follows:

*“Atoh1^En1-CKO^*mice developed severe ataxia, dystonia and tremor in the second to third week after birth and died shortly after weaning (P22-25), likely, because the motor phenotypes impairs food and water intake.”

7) The differential effects of caffeine on control and Atoh1^En1-CKO^ suggest differential roles of adenosine signaling. That would be very interesting particularly since adenosine is thought to contribute to respiratory roll-off during hypoxia. In any case, caffeine has been shown to increase baseline breathing and chemoreception in several animals including humans (PMID: 18703176) so we agree with the authors that it is very surprising caffeine decreased respiratory output in control mice: This unusual response could be due to the high baseline frequency. There are certainly also other explanations: caffeine also inhibits A2 receptors, which are excitatory so perhaps this is an A2 dependent effect? The authors should discuss these possibilities. Of course, the authors could also try selective A1 and A2 blockers to test this possibility, but we feel that this would exceed the scope of the present study.

We agree with the reviewers that the observed effects of caffeine on respiratory output is unexpected and that the mechanism underlying this result is an interesting question for future studies. We now discuss the difference between A1 and A2 signaling as follows:

“Caffeine treatment, along with glutamate and substance P, can stimulate endogenously slow respiratory rhythms in rats (Ruangkittisakul et al., 2010); this is thought to be the mechanism by which caffeine reduces apneas of prematurity in newborn infants (Aranda et al., 1977; Natarajan et al., 2007). […] Future studies with specific adenosine agonist and antagonist are needed to elucidate the specific effects of long-term caffeine treatment on respiratory control in neonatal mice.”

8) We would appreciate if the authors would also discuss whether they propose that PBC neurons are primary chemosensory neurons or a requisite component of the pathway.

We propose that PBC neurons are a requisite component of the pathway. We have clarified this in the Discussion as follows:

“Therefore, we propose that *Atoh1*-lineage PBC neurons are an essential component in the chemosensory pathways that relays information from peripheral, and perhaps central, chemosensors to enhance the firing of central chemosensitive and rhythmogenic neurons within the respiratory network.”

9) Previous evidence showed that CO_2_/H^+^-sensitive PBC neurons are glutamatergic and express calcitonin gene-related peptide (CGRP) (PMID: 29103805; PMID: 25424719). While we don't ask for additional experiments, it would be helpful to know if these markers overlap with Atoh1.

We thank the reviewers for suggesting this experiment. We have now included a figure (Figure 1—figure supplement 1) that shows the overlap of *Atoh1*-lineage neurons with the substance P receptor NK1R and neuropeptides CGRP and PACAP, that have been implicated in the hypercapnic and hypoxic respiratory response, respectively. We have discussed this in the text as follows:

“There are no previous reports that PBC neurons are intrinsically chemosensitive, so these neurons are likely activated by upstream neurons that are chemosensitive. […] Together, these results suggest that *Atoh1*-lineage neurons might be important for respiratory responses through signaling with one or both of these neuro peptides.”

10) A proposed role of CO_2_/H^+^-sensitive PBC neurons is maintenance of airway patency and there is evidence suggesting subsets of PBC neurons project to respiratory motor neurons (PMID: 25424719). However, results presented here (Figure 5) do not support this possibility. These differences should be discussed.

We are aware of the findings that PBC neurons might project to respiratory motor neurons. We think that the difference between our findings and the findings of previous studies are caused by the difference in neurons targeted (only *Atoh1*-lineage vs. the whole PBC population) and because of the projection mapping technique (anterograde mapping based on genetics vs. retrograde mapping based on local injections of Cholera toxin subunit B). We addressed this in the Discussion as follows:

“Lastly, our results confirm and extend previous work showing that lesioning the parabrachial nucleus impairs respiratory chemoresponses (Mizusawa et al., 1995; Song and Poon, 2008; 2009) and that the parabrachial nucleus projects to the ventrolateral medulla (Fulwiler and Saper, 1984; Holstege, 1988; Yokota et al., 2015). […] Therefore, we propose that *Atoh1*-lineage PBC neurons are an essential component in the chemosensory pathways that relays information from peripheral, and perhaps central, chemosensors to enhance the firing of central chemosensitive and rhythmogenic neurons within the respiratory network.”

11) The 100% neonatal death in Atoh1 null mice is very interesting. Did the authors observe whether the mice took a breath? Did they turn blue? Clarifying this would be very important for follow up studies.

We added this clarification in the text as follows:

“We have not caught any of these animals alive. Several *Atoh1^dCKO^* mice were found dead in the cage on the day of birth. These animals had turned blue and there was no visible milk spot, suggesting that they died of respiratory failure before drinking any milk.”